# Coordination of transporter, cargo, and membrane properties during non-vesicular lipid transport
Alena Ballekova [1,3] ✉, Andrea Eisenreichova[1,3], Bartosz Różycki [2], Evzen Boura [1] &
Jana Humpolickova [1] ✉

Homeostasis of cellular membranes is maintained by fine-tuning their lipid composition. Yeast lipid transporter Osh6, belonging to the oxysterol-binding protein-related proteins family, was found to participate in the transport of phosphatidylserine (PS). PS synthesized in the endoplasmic reticulum is delivered to the plasma membrane, where it is exchanged for phosphatidylinositol 4-phosphate (PI4P). PI4P provides the driving force for the directed PS transport against its concentration gradient. In this study, we employed an in vitro approach to reconstitute the transport process into the minimalistic system of large unilamellar vesicles to reveal its fundamental biophysical determinants. Our study draws a comprehensive portrait of the interplay between the structure and dynamics of Osh6, the carried cargo lipid, and the physical properties of the involved membranes, with particular attention to the presence of charged lipids and to membrane fluidity. Specifically, we address the role of the cargo lipid, which, by occupying the transporter, imposes changes in its dynamics and, consequently, predisposes the cargo to disembark in the correct target membrane.

Membranes of specific cellular organelles significantly differ in fluidity, thickness, and lipid composition[1]. This is an essential feature for their functionality and identity. Most phospholipids are synthesized in the endoplasmic reticulum (ER) and must be transported to their final destination. This process is critical and must be ingeniously orchestrated.

The lipid molecules are transported either by vesicles that bud from the source membrane and coalesce with the target membrane[2], or by non-vesicular pathways using lipid transfer proteins (LTPs)[3–5]. It has been shown that LTPs rectify the inter organelle flux[5], are involved in organelle biogenesis, membrane repair, and lipid rearrangement[6]. The bridge-like LTPs were shown to have a significant role in the lipid sorting pathways[6].

The yeast transporter of phosphatidylserine (PS), oxysterol-binding protein (OSBP) homolog Osh6[7–9], belongs to the family of transfer proteins known as OSBP-related proteins (ORPs). Osh6 is a soluble protein localized between the ER and the plasma membrane (PM)[10], where it associates with the membrane tether Ist2[11]. It transports PS from the ER to the PM by exchanging it for phosphatidylinositol 4-monophosphate (PI4P)[12]. By this process, PI4P is released in the ER membrane, where it is dephosphorylated by Sac1 phosphatase[13,14]. The energy required for the counter-gradient

transport of PS is spent on maintaining the PI4P pool in the PM, which consists of the PI transport and its subsequent phosphorylation by phosphatidylinositol 4-kinase.

In our previous work[15], we showed that PS can be transported spontaneously down its gradient without the need for exchange. However, in cells, the transport occurs against the gradient, which requires a deeper mechanistic understanding that includes the different affinities of Osh6 and the individual transported lipids. PI4P has a higher affinity for the Osh6 binding pocket than PS[7] and can replace the PS molecules, inhibiting the along-gradient transport. To load PS again from the PS-donating membrane, Sac1 is required on that membrane to dephosphorylate the incoming PI4P. In this manuscript, we will refer to PS and PI4P as cargo lipids, even though in our previous work, we have seen that phosphatidylinositol-4,5-bisphosphate (PIP2) can also be shuffled by Osh6 between membranes[15] and others observed that Osh6 can associate with phosphatidylglycerol (PG)[8].

Since lipid transport involves not only the interaction between the cargo lipid and the protein, but also the interaction between the entire membrane and the protein and the interaction between the transported lipid and the involved membranes, we focus on the role of the membrane in the

[1]Institute of Organic Chemistry and Biochemistry of the Czech Academy of Sciences, Prague, Czechia. [2]Institute of Physics, Polish Academy of Sciences, Warsaw, Poland. [3]These authors contributed equally: Alena Ballekova, Andrea Eisenreichova. ✉e-mail: alena.ballekova@uochb.cas.cz; jana.humpolickova@uochb.cas.cz

process. Specifically, we investigate the role of certain membrane features, such as presence of charged lipids and fluidity, on Osh6-mediated lipid transport and relate our findings to the electrostatic properties and dynamics of the transporter. It has been previously shown that Osh6 interacts differently with neutral and charged membranes, and that the interaction is affected by the presence of cargo in the lipid binding pocket[16]. We further elaborate on this by examining the process of cargo extraction and deposition to the target membrane. Our results demonstrate that the kinetics of extraction depend on both the cargo lipid and the characteristics of the entire membrane. Additionally, the cargo lipid influences the interaction of Osh6 with the membrane and determines whether the charged membrane facilitates the release of cargo or not.

## Results

In the following section, we will examine how membrane properties, particularly the presence of charged lipids and membrane fluidity, influence the extraction and release of individual lipid cargoes. Furthermore, we will compare the membrane-binding properties of Osh6 in both its unbound form and when occupied by its ligands. Using this knowledge, we will demonstrate that Osh6 transports PS to membranes enriched in PS and containing PI4P, by effectively exchanging PS for PI4P. This provides a fundamental biophysical understanding of the processes occurring in cells.

### Extraction and release

To understand transport as such, we discriminate between two distinct processes: lipid extraction and release. Even though temporally they are not totally separated, we think that dealing with them separately helps grasp the regulatory aspects involved in each of them independently. To do that we employ fluorescence cross-correlation spectroscopy (FCCS) and leverage our understanding of the behavior of the lipid biosensor. FCCS is microscopy technique that deals with single fluorescent particles that in our case are fluorescently labeled either liposomes (large unilamelar vesicles - LUVs) or lipid biosensors.

Fig. 1A shows the scheme of the experiment. The two overlapping excitation laser beams are focused into a single spot. The spot represents an observation cuvette from which the signals of individual fluorescent species are collected. The fluctuations in the green and red signals (Fig. 1B) result from the passage of diffusing particles and thus carry information on the dynamics and number of green and red species. Additionally, statistically significant temporal co-occurrences of fluctuations in both emission channels refer to the number of particles that carry both fluorophores. The intensity traces (Fig. 1B) are evaluated by temporal auto- and cross-

correlation functions: $G_G$, $G_R$, and $G_{cc}$:

$$G_{G,R}(\tau) = \frac{\langle \delta I_{G,R}(t)\delta I_{G,R}(t+\tau)\rangle}{\langle I_{G,R}\rangle^2}, \quad G_{cc}(\tau) = \frac{\langle \delta I_G(t)\delta I_R(t+\tau)\rangle}{\langle I_G\rangle\langle I_R\rangle}, \quad (1)$$

where $I_{G,R}$ is the fluorescence intensity in the green and the red channel, respectively. The square brackets stand for the temporal average. $\delta I$ represents the fluctuation in fluorescence intensity. The amplitude $G_{cc}(0)$ increases with the frequency of the co-occurring bursts in both the channels (arrows in Fig. 1B) and thus is related to the degree of bound biosensor and consequently to the level of the sensed lipid. To avoid effects of inaccuracies in LUV labeling by lipid tracer DiD, the read-out parameter used throughout the manuscript is $G_{cc}(0)/G_R(0)$, in text referred as $G_{cc}(0)$ for simplicity.

In all our transport experiments, we monitor the temporal drop in the fraction of the lipid that is being transported from the labeled LUVs. The biosensors used in the manuscript dynamically sense the lipid of interest only within a certain range of concentrations, above this range, the response gets saturated. The concentration of cargo lipid always exceeds the concentration of the biosensor. The cargo that is available at the surface of LUVs is extracted and by shift of the binding equilibrium, the biosensor detaches. The illustrative dependence of $G_{cc}(0)$ on the fraction of the sensed lipid in labeled liposomes is shown in Fig. 1C. The experiments are performed in two distinct modes: (i) extraction of the cargo from the donor membrane by Osh6, and (ii) release of the cargo to the acceptor membrane.

In the case of extraction, the accessible fraction of the cargo lipid is low and equal to the concentration of Osh6. The change in $G_{cc}(0)$ upon Osh6 addition is instantaneous, as the biosensor is very sensitive to the decrease of the lipid of interest (Fig. 1C, red solid line).

In the case of release, the amount of lipid in LUVs is selected so that the response of the biosensor is already saturated, while the concentration of added Osh6 remains the same as in the case of extraction, i.e., it has capacity to extract only a fraction of the cargo lipid. Since the response of the biosensor is saturated, the drop in $G_{cc}(0)$ corresponding to the initial extraction is small (Fig. 1C, blue dotted curve). At this point, Osh6 is loaded with cargo; thus, the further drop in $G_{cc}(0)$ predominantly refers to the release of the cargo lipid (Fig. 1C, red dotted curve) to unlabeled LUVs present in the system.

### PS extraction and release

LUVs that contain PS and a fluorescent lipid tracer, DiD, are prepared. For PS visualization, the PS biosensor, C2 domain of Lactadherin C fused to CFP (C2$_{Lact}$-CFP)[17–19] is added to the LUVs. As a result, the LUVs are double-

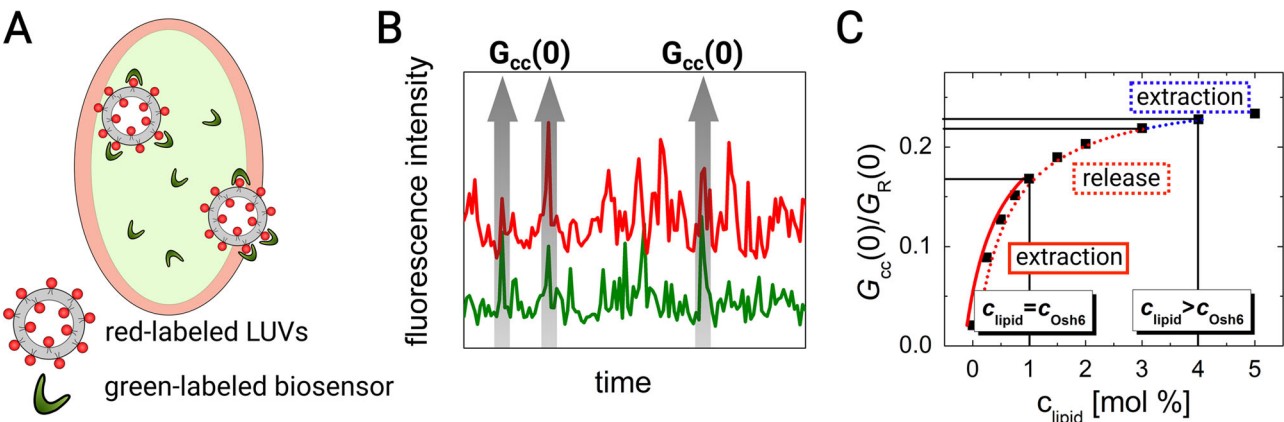

**Fig. 1 | Explanation of the assay. A** Schema of the two overlapping laser foci of the confocal microscope, which form a light cuvette. Transition of the fluorescent LUVs and of the lipid biosensors are visualized as signal fluctuations. **B** Example of fluctuating green and red signals with a high level of cross-correlation $G_{cc}(0)$. **C** FCCS read-out, $G_{cc}(0)/G_R(0)$, as a function of cargo lipid in labeled LUVs (black squares). The two regimes of the transport experiments: (i) extraction (red solid line), and (ii) release (red dotted line). The blue dotted line stands for the initial change in $G_{cc}(0)/G_R(0)$ due to extraction immediately upon Osh6 addition before the release regime starts, which occurs upon the Osh6 addition.

**Fig. 2 | PS extraction assay. A** Scheme of the assay. The number of accessible PS ($n_{PS}$(acc.)) equals the amount of Osh6. The FCCS read-out parameter ($G_{cc}(0)/G_R(0)$, simply $G_{cc}$) monitoring the mutual motion of DiD and C2-Lact fused to CFP drops upon Osh6 addition. **B** Scheme of the assay when the PS donating LUVs are in presence of other, PS-free LUVs either not bearing a competitive ligand (1)—PS extraction occurs or bearing a competitive ligand (2)—PS extraction is compromised. **C** Dependence of the $G_{cc}$ read-out of C2$_{Lact}$-CFP biosensor in our experimental system. The red line depicts the region of the calibration curve where extraction can be observed. **D** Temporal drop in $G_{cc}$ during the PS extraction. The extraction from LUVs containing 1% PS was carried out in the absence of other, PS-free LUVs (black), or in the presence of other, PS-free LUVs differing in the lipid composition: POPC (red), POPC/POPE (blue), POPC/POPG (green), POPC/POPA (orange), POPC/soy PI (dark yellow), POPC/PI4P (violet), POPC/PIP$_2$ (magenta). **E** Evaluation of extraction rates depending on the lipid added to the LUVs B. Composition of the experiment in 1D, E: $c_{POPS}$ = 500 nM, $c_{Osh6}$ = 250 nM, $c_{C2lact-CFP}$ = 50 nM, $c_{total\ lipids\ LUV\ A}$ = 50 µM, $c_{total\ lipids\ LUV\ B}$ = 200 µM. All error bars represent the standard error of the mean, n = 10 measurements. The p-values were obtained from the two-sample t-test.

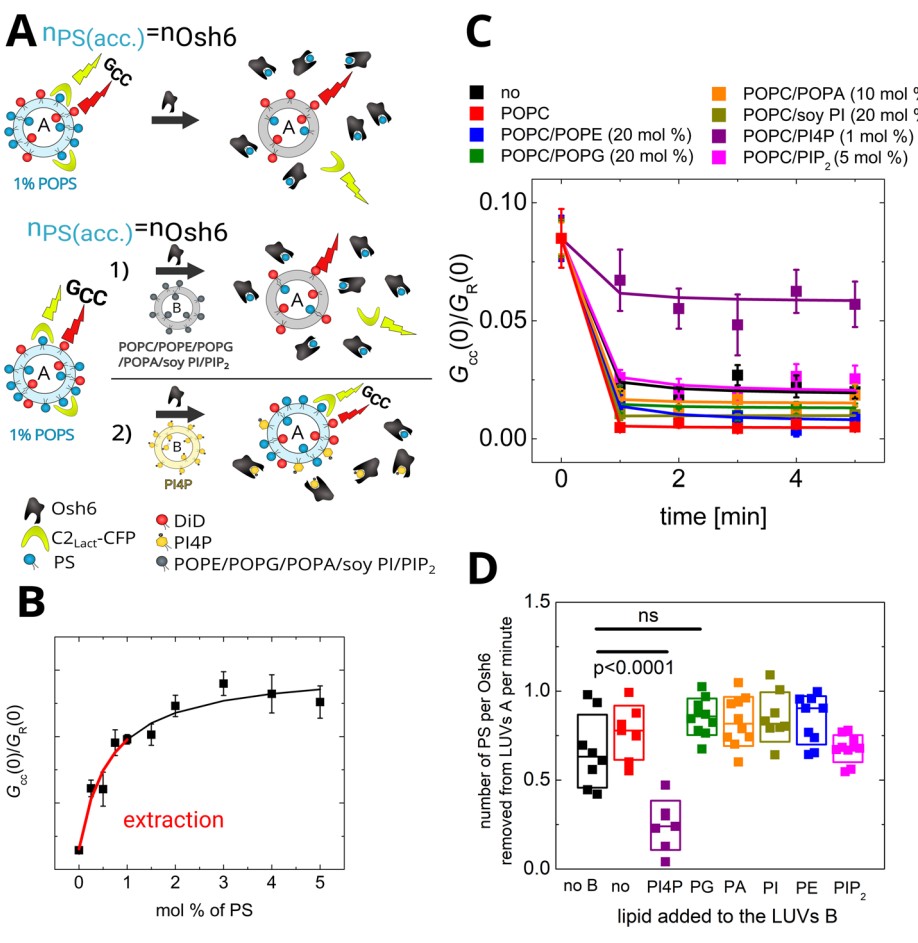

labeled with both CFP and DiD, which leads to a high level of cross-correlation between the CFP and the DiD signal, as both fluorophores move together. Upon addition of Osh6, PS is extracted from the LUVs (type A) and can be further deposited to a different membrane (LUVs type B). At this point, the biosensor detaches from the donor membrane, and the cross-correlation amplitude ($G_{cc}(0)$) drops down. We conducted the experiment to either examine the kinetics of the extraction (i) or the release of the cargo to a different membrane (ii).

(i) When the level of accessible PS in LUVs is stoichiometrically equivalent to the amount of Osh6, PS extraction from the donor membrane is observed upon Osh6 addition (Fig. 2A–C). The purpose of this experiment was to identify other potential ligands of Osh6. Therefore, apart from the PS donating LUVs (LUVs A) also LUVs B containing high excess of other cargo candidates were examined with respect to their competition for the Osh6 binding pocket (Fig. 2B). The extraction occurs instantaneously and almost irrespective of whether other types of LUVs (LUVs B) are present in the system (Fig. 2D, E). The only exception is for LUVs that contain other Osh6 ligands, such as PI4P, which compromises PS extraction by competing for the binding site. Even though PIP$_2$ and PG were previously shown to either be transported by Osh6[15], or associate with it[8], in this experiment none of them significantly interfere with the extraction. Their binding affinity may be significantly lower compared to the main cargo molecules, such as PS and PI4P. Our experiment thus does not show any other significantly binding ligands of Osh6.

During the time span of the experiment with the population of LUVs B, it is probable that eventually some PS will be released to the PS-accepting membrane. We analyzed the diffusion properties of C2$_{lact}$-CFP before adding Osh6 and at the last time point of the experiment. Fast diffusion of the biosensor (i.e., not bound to the slowly diffusing liposomes) in all cases

where $G_{cc}(0)$ dropped to almost zero (Fig. 2D) suggests that the majority of the extracted PS remained in the transporter. The corresponding curves are depicted in Fig. S1. The transport itself takes a longer time and is discussed further.

Additionally, to ensure that the decrease in $G_{cc}(0)$ is not solely due to the competition between C2$_{lact}$-CFP and Osh6 for cargo binding, we examined the extraction of the non-extractable lipid diphytanoylPS (phPS). As shown in Fig. S2, no change in $G_{cc}(0)$ occurs upon the addition of Osh6, even at a fourfold excess, suggesting that the observed decrease in $G_{cc}(0)$ in Fig. 2D is indeed caused by the extraction of PS, rather than the displacement of the biosensor by Osh6.

(ii) When the level of accessible PS exceeds the level of Osh6 (Fig. 3A, B), the addition of Osh6 causes PS extraction, followed by its deposition to the acceptor membrane. The level of observable extraction is minimized (Fig. 3C—blue). Experimentally this can be accessed, when no other LUVs are available (Fig. 3D–F black squares). However, if acceptor LUVs are present, the drop in $G_{cc}(0)$ can be almost solely attributed to ligand release to the acceptor membrane (as the PS extraction is fast, i.e., occurs during the first minute upon Osh6 addition, which is a temporal resolution of our experiment) (Fig. 3B, D–I).

Figure 3 sheds light on two membrane determinants that significantly affect the release of PS to the accepting membrane: (i) presence of charged lipids and (ii) membrane fluidity. While the drop in $G_{cc}(0)$ upon sole extraction of PS from LUVs containing 4 mol% PS is small (Fig. 3D–F black squares), the presence of acceptor LUVs allows for further flow of PS towards the accepting membrane. The extent of the PS release is highly influenced by the charged lipids in the accepting membrane. Lipid head groups of PG, phosphatidic acid (PA), and phosphatidylinositol (PI) significantly improve the release compared to the neutral membrane. In contrast, the PI4P head group inhibits PS transport, which is in good

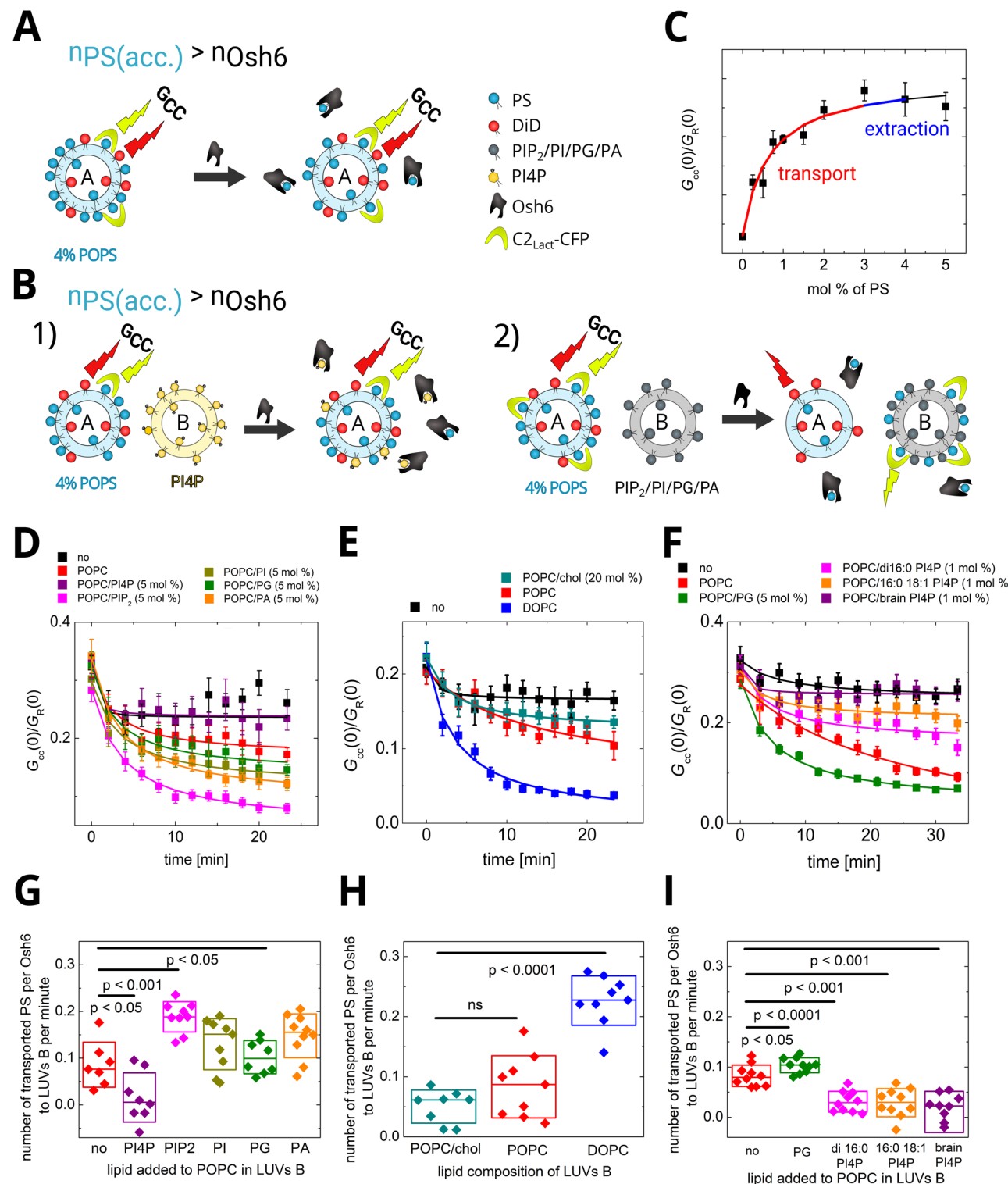

agreement with the previous finding that PI4P blocks the binding pocket of Osh6[15].

We have also paid specific attention to the presence of $PIP_2$ in the acceptor membrane, as $PIP_2$ was shown to have a great impact on the transport of dehydroergosterol (DHE) by Osh4[10]. In the case of Osh6-mediated transport of PS, we also observe a significant impact of $PIP_2$ on the level of PS transport. However, considering the higher, locally concentrated charge of the $PIP_2$ molecule (net charge –4 at neutral pH[20]), its effect is comparable to the impact of other charged membranes (PI, PG, PA—net charge of all around –1[21]).

Another important aspect of the PS release to the acceptor membrane is the membrane fluidity. The more fluid the acceptor membrane is, the better it accepts the cargo (Fig. 3E). While cholesterol containing 1-palmitoyl-2-oleoyl-glycero-3-phosphocholine (POPC) almost does not accept the POPS cargo, the membrane composed of highly fluid 1,2-dioleoyl-glycero-3 phosphocholine (DOPC) absorbs the cargo readily.

**Fig. 3 | PS transport assay. A** Scheme of the assay. The number of accessible PS ($n_{PS}(acc.)$) is higher than the amount of Osh6. The FCCS read-out, $G_{cc}$, monitoring the mutual motion of DiD and C2$_{Lact}$ fused to CFP remains almost unchanged upon the Osh6 addition as the relative change of PS in the donor LUVs is small and the biosensor's response is saturated. **B** Scheme of the assay when the PS donating LUVs are in presence of acceptor LUVs either containing a competitive cargo (1) – Osh6 is blocked by the cargo, or not containing a competitive cargo (2) – transport to the acceptor LUVs occurs. **C** Dependence of the $G_{cc}$ read-out of C2$_{Lact}$ biosensor in our experimental system. The blue and red lines show the concentration regions of extraction and transport, respectively. **D** Role of charged lipids in the PS release. Temporal drop in $G_{cc}$ during the PS transport. No PS accepting LUVs were added (black), PS accepting LUVs were composed of: POPC (red), POPC/PI4P (violet), POPC/PIP$_2$ (magenta), POPC/PI (dark yellow), POPC/PG (green), POPC/PA

(orange). **E** Role of membrane fluidity in the PS release. Temporal drop in $G_{cc}$ during the PS transport. No PS accepting LUVs were added (black), PS accepting LUVs were composed of: POPC/cholesterol (cyan), POPC (red), DOPC (blue). **F** Role of aliphatic chains of PI4P in the PS release. Temporal drop in $G_{cc}$ during the PS transport. No PS accepting LUVs were added (black), PS accepting LUVs were composed of: POPC (red), POPC/POPG (green), POPC/di16:0 PI4P (magenta), POPC/16:0 18:1 PI4P (orange), POPC/brain PI4P (violet). **G–I** Evaluation of the transport rates corresponding to observations depicted in (**D–F**), respectively. Composition of the experiment in 2D-I: $c_{POPS} = 2\ \mu M$, $c_{Osh6} = 250\ nM$, $c_{C2lact-CFP} = 50\ nM$, $c_{total\ lipids\ LUV\ A} = 50\ \mu M$, $c_{total\ lipids\ LUV\ B} = 200\ \mu M$. All error bars represent the standard error of the mean, n = 10 measurements. The p-values were obtained from the two-sample t-test.

The PI4P used in most experiments presented throughout this manuscript is the mixture from porcine brain with high levels of poly-unsaturated fatty acids, such as arachidonic acid. However, these acyl chains are not present in yeast, where saturated and monounsaturated chains prevail[22]. The binding affinity of yeast variants of PI4P to Osh6 was shown to be lower[23], which also changes the behavior of the ligand with respect to Osh6-mediated transport. In Fig. 3F, we draw attention to the inhibition of PS transport along the gradient by di16:0 PI4P (magenta), 16:0 18:1 PI4P (orange, the most abundant in yeast), and by the porcine brain mixture (violet). Consistent with their binding affinities, the three ligands demonstrate different degrees of inhibition of PS transport. The stronger the ligand, the more efficiently it halts the process. However, none of the tested species is inactive in inhibition, or even accelerates the process. For comparison, the acceleration caused by the addition of 5 mol% of negatively charged PG is also shown (Fig. 3F, green squares).

### PI4P extraction and release
We have shown that PI4P can replace PS in the Osh6 binding pocket and inhibit the along-gradient PS transport. To transport PS against the gradient continuously, PI4P eventually must be released to the PI4P accepting membrane, i.e., to the ER in cells. We have shown that Sac1-assisted dephosphorylation of PI4P keeps shifting the equilibrium between PI4P bound to Osh6 and PI4P in the accepting membrane[15]. Despite that, before being dephosphorylated, PI4P must escape the binding pocket while Osh6 gets in contact with the membrane. Here, we focus on the membrane features that facilitate the PI4P transport. Similar to our examination of PS, both the PI4P uptake and release are addressed by employing FCCS in a similar setup. The PI4P donating LUVs are labeled with DiD, and PI4P is sensed by SidC-Atto488, so at the beginning, the double-labeled vesicles show high cross-correlation ($G_{cc}$) that drops while PI4P is extracted from the donor membrane.

Figure 4 addresses the kinetics of PI4P uptake, when the level of extractable PI4P is equivalent to the amount of Osh6 used (Fig. 4A–C). Figure 4D (black curve), E show that indeed, the entire PI4P can be extracted upon Osh6 addition. It is also worth noticing that PI4P uptake is a much slower process than the uptake of PS. While extracting the PI4P pool requires almost 10 min, the kinetics of PS extraction cannot be captured within the temporal resolution of our experiment; all available PS is extracted in less than a minute (compare Figs. 2D and 4D). The presence of other LUVs did not cause significant changes in the PI4P extraction kinetics except for LUVs containing excess PS, which competes with PI4P for the binding pocket (Fig. 4B, D, E). To prove that the PI4P was not transported during the extraction, Fig. S1 shows diffusion properties of SidC-Atto488 in those cases where the extraction occurred, i.e., $G_{cc}(0)$ dropped to zero (Fig. 4D). The fast diffusion of the biosensor refers to the fact that PI4P was not released to the acceptor membrane.

In the following experiments, we focused on the PI4P release to the acceptor membrane (Fig. 5A–C). The level of PI4P was set to 3 mol%, i.e., above the capacity of Osh6, so that even if each Osh6 extracted a PI4P molecule, almost no drop in $G_{cc}$ was observed as the response of the biosensor under these conditions was saturated (Fig. 5D, black squares).

We then added other LUVs to the system to enable Osh6 to deposit the PI4P cargo to the acceptor membrane. The excess of PS cargo in the acceptor membrane, which can quickly bind to the Osh6 binding pocket, allows for better PI4P release (Fig. 5D, red hollow squares). In contrast, membranes containing the negatively charged lipids other than cargo (Fig. 5E, orange and green squares) or contain diphytanoylPS (phPS) (Fig. 5D, red solid circles), the lipid with a PS head group and methylated fatty acid chains, which makes it unextractable, reduce the drop of PI4P to the acceptor membrane.

PI4P is obviously better accepted by membranes that are more fluid (DOPC rather than POPC). Also, addition of diacylglycerol (DAG) to the PC bilayer promotes PI4P release (Fig. 5F, dotted curves). Figure 5F also shows that if the target membrane was PM-like (negatively charged—10 mol% POPS, and rigid—25 mol% cholesterol), the transport is inhibited totally.

The effects observed in Figs. 3D–I and 5D–G, i.e., the impacts of membrane properties on PS and PI4P release from Osh6, respectively, reflect: (i) the thermodynamic partitioning of the cargo lipid between two coexisting membranes (the drop in $G_{cc}(0)$ at infinite time), governed by lipid-membrane interaction, and (ii) the interaction of the loaded protein with specific membrane properties (the rate of the drop before equilibrium is reached), i.e., protein-membrane interaction. In summary, the data show that transporting PS to fluid and charged membranes is advantageous. PS appears to be better accommodated in fluid membranes lacking cholesterol and in membranes bearing charge. For the release step itself, the interaction of the weakly closed lid (compared to PI4P) with acidic and fluid membranes seems to be beneficial.

The main difference between the release of PS and PI4P is that PI4P release is not enhanced by negatively charged lipids other than the cargo itself. Therefore, membrane fluidity may have a more decisive impact on its release. Here, the DOPC membrane with DAG represents the fluidity characteristics of the ER membrane, contrasting with the POPC/POPS/cholesterol membrane, which is more rigid and would more closely resemble the PM.

Our extraction results confirm that PI4P is a stronger binder to Osh6 than PS. The violet squares in Figs. 2D and 4D represent the inhibition of PS extraction by PI4P and PI4P extraction by PS, respectively. A similar level of inhibition was achieved with 1 mol% of PI4P in the case of PS (1 mol%) extraction, and 20 mol% of PS in the case of PI4P (1 mol%) extraction. Consistent with the literature, molecular simulations predict a higher binding energy for PI4P compared to PS[7], and the melting point of Osh6 is also higher for PI4P than for PS[22]. This explains why PI4P release is accelerated by an excess of PS, which replaces PI4P in the Osh6 binding pocket due to its concentration. Additionally, during PS transport to PI4P-containing LUVs, the replacement of PS by PI4P halts the cyclic process of PS extraction and release, as the transporter becomes occupied by a stronger ligand and is unavailable for another cycle. Notably, after releasing PI4P from the lipid transfer domain to acceptor organelle, ER, metabolic conversion of PI4P to PI occurs to continue the cycle within cells[7,24].

### Membrane binding of Osh6
LTPs, specifically their transfer domains, are designed to extract cargo when empty (i.e., bind to the membrane) and leave the membrane when loaded

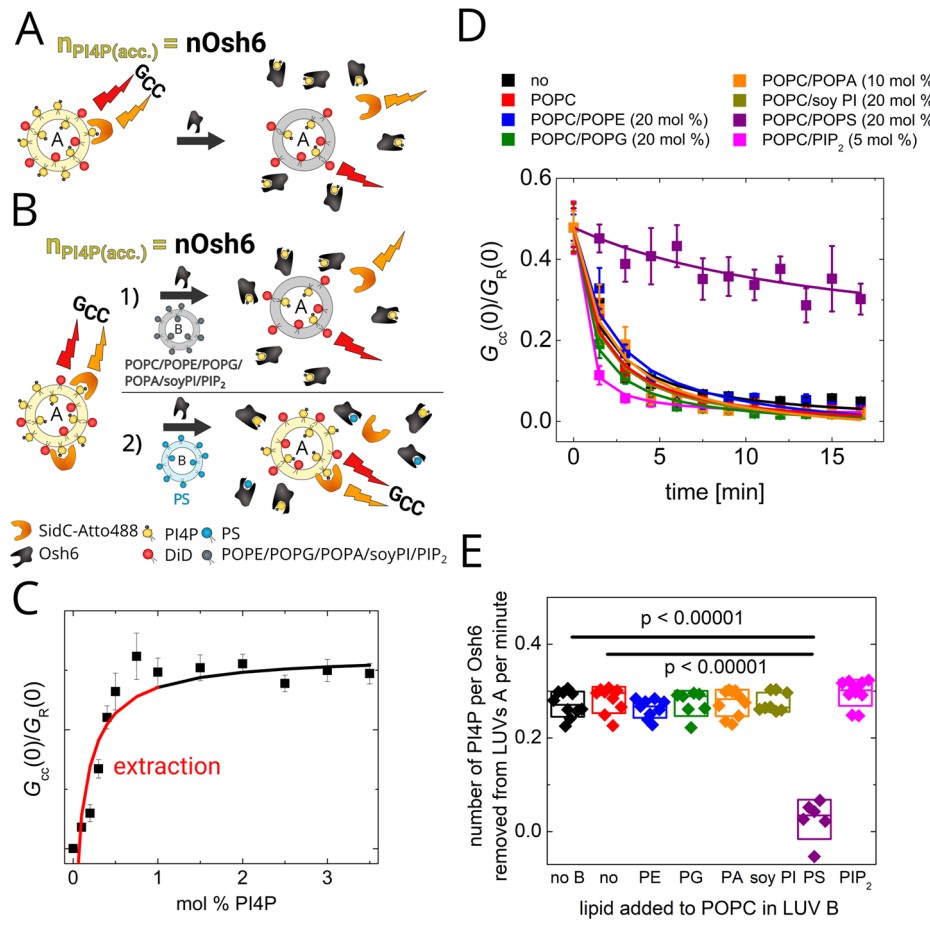

**Fig. 4 | PI4P extraction assay. A** Scheme of the assay. The amount of accessible PI4P ($n_{\text{PI4P}}$(acc.)) equals the amount of Osh6. The FCCS read-out, $G_{cc}$, monitoring the mutual motion of DiD and SidC-Atto488 disappears upon the Osh6 addition. **B** Scheme of the assay when other, PI4P-free LUVs (marked as B) are present during the extraction. The LUVs B either do not contain the competing cargo (1)—the extraction is not affected or contain the competing cargo (2)—the extraction is compromised. **C** Dependence of the $G_{cc}$ read-out of SidC-Atto488 biosensor in our experimental system. The red line depicts the concentration region of PI4P where the extraction takes place. **D** Kinetics of the PI4P extraction from the PI4P containing LUVs when no other LUVs are present (black), and at presence of other, PI4P-free LUVs of various compositions: POPC (red), POPC/POPE (blue), POPC/POPG (green), POPC/POPA (orange), POPC/soy PI (dark yellow), POPC/POPS (violet), POPC/PIP$_2$ (magenta). **E** PI4P extraction rates for each composition of LUVs B. Composition of the experiment in 3D, E: $c_{\text{PI4P}}$ = 500 nM, $c_{\text{Osh6}}$ = 250 nM, $c_{\text{SidC-Atto488}}$ = 100 nM, $c_{\text{total lipids LUV A}}$ = 50 μM, $c_{\text{total lipids LUV B}}$ = 200 μM. All error bars represent the standard error of the mean, n = 10 measurements. The p-values were obtained from the two-sample t-test.

with cargo. Generally, these proteins are not strong membrane binders, but they do transiently interact with the membrane at some point. In the following section, we will compare the binding of empty proteins and cargo-loaded proteins to different types of membranes, with the aim of better understanding the role of charged lipids in lipid transfer.

A cysteine-less mutant of Osh6 was prepared with a single amino acid residue mutated to cysteine for specific labeling. The transport behavior of the mutant was identical to that of the wild-type protein. It was then labeled with Atto488, and its binding to various types of membranes was explored. Osh6-Atto488 was added to LUVs of defined composition that contained DiD, the fluorescence marker. In addition, Osh6-Atto488 was pre-incubated with either PI4P or PS-containing LUVs long enough to saturate the transporter with cargo (Fig. 6A). Using FCCS, the temporal cross-correlation between the green and red signals was monitored.

Figure 6B shows the amplitudes of the cross-correlation curves between the Osh6-Atto488 and the DiD signals, which relate to the interaction between the protein and the membrane. The higher the amplitude is, the more labeled Osh6 molecules are attached to the membrane. First, the interaction of the protein with various LUVs A was monitored (Fig. 6B, black dots). Keeping in mind that PS and PI4P are the main cargo lipids, and that the membrane containing these lipids will also provide Osh6 with the cargo, we can conclude, in agreement with the literature[16], that Osh6 binds weakly to the cargo-donating membrane. Slightly more pronounced adhesion to the neutral membrane is observed. Osh6 adheres better to membranes that contain charged lipids such as PG and PA and best to membranes containing lipids with negatively charged headgroups and with phytanoyl chains, such as diphytanoylPG (phPG), diphytanoylPA (phPA), or diphytanoylPS (phPS). Osh6 recognizes phPS as a ligand but, due to the methylated lipid chains, cannot accommodate it and pull it out from the

membrane. phPG and phPA do not contain a ligand headgroup, yet presence of both in LUVs A increases Osh6 binding. The difference in the Osh6 binding between the phytanoyl tails in phPG and phPA and the palmitoyl and oleoyl chains in POPG and POPA, respectively, can perhaps be attributed to better exposure of the charge due to lipid packing defects[25]. The binding is not much affected by the charge of PI, which may be shielded by the inositol ring. Also, PIP$_2$ is a highly negatively charged lipid that does not significantly increase the membrane adhesion of Osh6, but it has to be kept in mind that PIP$_2$ was also identified as a weakly binding cargo of Osh6[15].

Second, to learn about the interaction of PS- or PI4P-loaded Osh6, we pre-incubated Osh6-Atto488 with label-free LUVs B containing either PS or PI4P. We then added the mixture to the solution of labeled LUVs (Fig. 6B, blue and yellow dots). Adhesion to all types of labeled cargo-free LUVs dropped, which was more pronounced for the PI4P-loaded Osh6. Figure 6C shows the temporal cross-correlation curves for the LUVs containing lipids with phytanoyl tails: phPS (black lines), phPG (red lines), and phPA (blue lines). The solid, dotted, and dashed lines show the interaction of empty, PS-, and PI4P-loaded Osh6, respectively.

Jointly, these results clearly suggest that the cargo changes the protein-membrane interaction, specifically, that the PS release from the binding pocket is charge-dependent compared to the PI4P release.

## Role of the cargo and charged lipids in membrane distinguishing

We have demonstrated that with membranes containing charged lipids, the empty Osh6 associates more strongly than Osh6 that has been incubated either with PS- or PI4P-containing LUVs and thus contains the appropriate cargo, consistent with previous findings[16]. Moreover, in the case of membranes containing charged diphytanoyl lipids (phPG and phPA), there is even a significant difference between the PS- and PI4P-loaded Osh6 (Fig. 6).

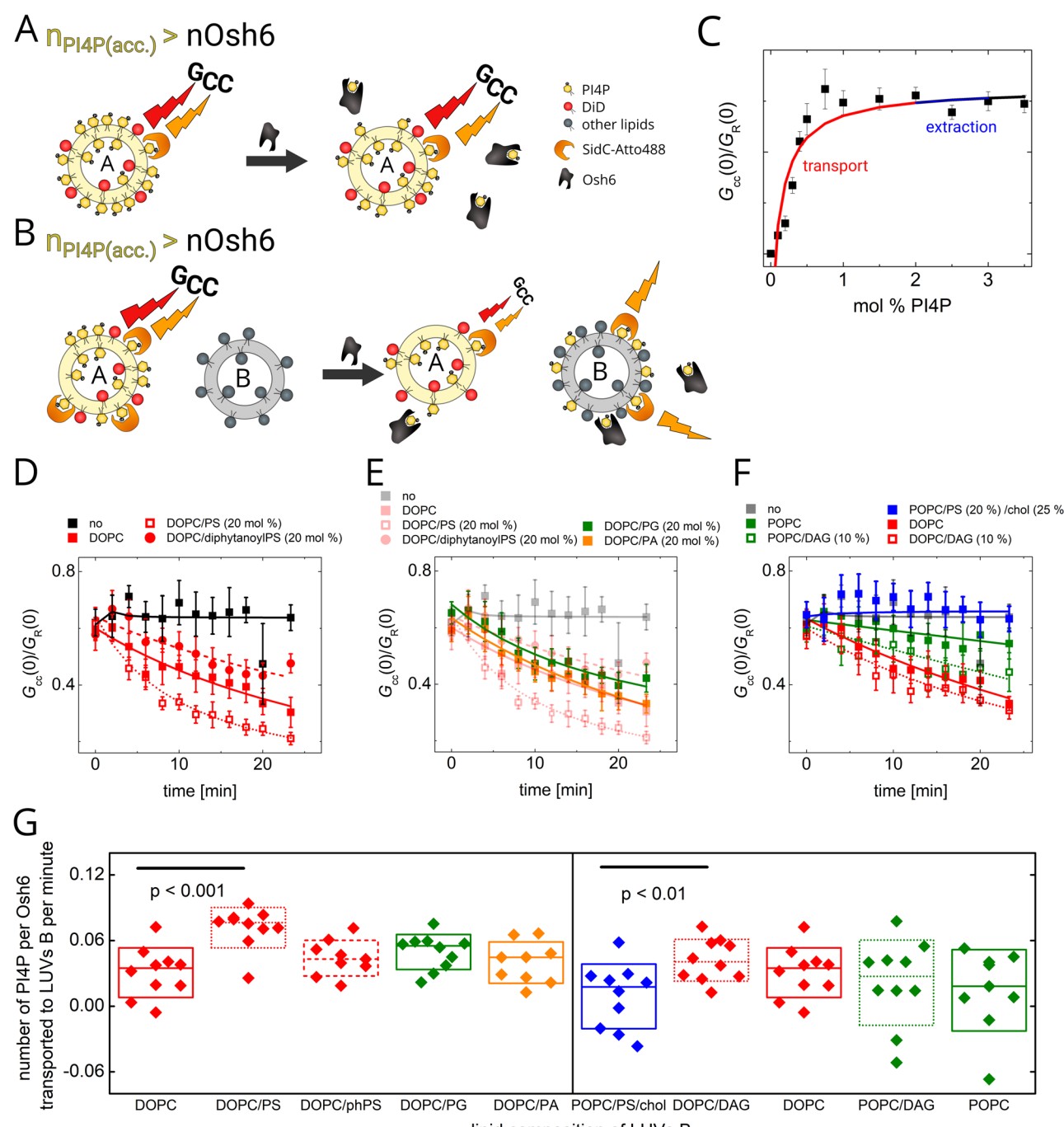

**Fig. 5 | PI4P transport assay. A** Scheme of the assay. The amount of accessible PI4P ($n_{PI4P}$(acc.)) exceeds the amount of Osh6. The FCCS read-out, $G_{cc}$, does not drop upon the Osh6 addition as the relative change of PI4P level in the PI4P donating LUVs is small and the biosensor's response is saturated. **B** Scheme of the assay when also PI4P accepting LUVs are present in the system. The FCCS read-out, $G_{cc}$, drops as the PI4P accepting LUVs allow for the PI4P deposition and continuation of the transport. **C** Dependence of the FCCS read-out, $G_{cc}$, of SidC-Atto488 biosensor in our experimental system. The blue and red lines depict the PI4P concentration regions where the extraction and the transport can be observed, respectively. **D** PI4P transport to LUVs with competing ligand. Kinetics of the PI4P transport to the PI4P accepting LUVs composed of DOPC (red solid squares), DOPC/PS (red hollow squares), DOPC/diphytanoylPS (red solid circles). The experiment with no accepting LUVs is depicted by black solid squares. **E** Effect of charged lipids on the PI4P release. The PI4P accepting LUVs were composed of DOPC/PG (green) and DOPC/PA (orange). The data from the (**D**) (partially transparent) are shown for comparison. **F** Effect of fluidity on the PI4P release. The PI4P accepting LUVs were composed of POPC (green solid squares), POPC/DAG (green hollow squares), DOPC (red solid squares), DOPC/DAG (red hollow squares), and POPC/POPS (10 mol %)/cholesterol (25 mol%)—PM-like composition (blue solid squares). Experiment without the accepting LUVs is shown for comparison (gray squares). **G** PI4P transport rates for each composition of LUVs B. Composition of the experiment in 4D–F: $c_{PI4P}$ = 1.5 µM, $c_{Osh6}$ = 250 nM, $c_{SidC-Atto488}$ = 100 nM, $c_{total\ lipids\ LUV\ A}$ = 50 µM, $c_{total\ lipids\ LUV\ B}$ = 200 µM. All error bars represent the standard error of the mean, n = 10 measurements. The p-values were obtained from the two-sample t-test.

Also, the knowledge on extraction kinetics raises questions dealing with the situation in cells. For example, it is not clear how PI4P gets extracted from the PS rich plasma membrane if the PS extraction is a much faster process than the extraction of PI4P. Motivated by this discrepancy, we

examined the extraction of PI4P from membranes containing high level of PS (20 mol%) using the assay shown in Fig. 4.

To replicate the conditions at the plasma membrane (PM) surface, where Osh6 transports PS from the ER and is tasked with delivering the

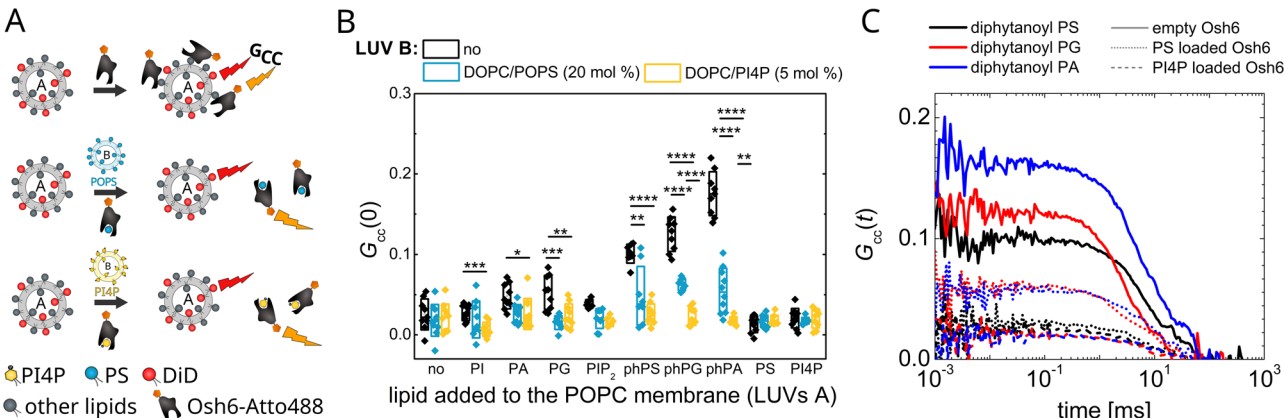

**Fig. 6 | Cargo effect on membrane binding. A** Scheme of the experiment. The DiDlabeled LUVs A of various compositions were mixed with Osh6-Atto488 that was either empty or pre-incubated with PS- or PI4P-containing LUVs B. The binding was observed as the FCCS read-out, $G_{cc}$, corresponding to the mutual motion of DiD and Atto488. **B** The FCCS read-out, $G_{cc}$, monitoring binding Osh6-Atto488 to membranes of various lipid composition (LUVs A) (POPC membrane contained 5 mol% of PI4P, or PIP$_2$, or 20 mol% of other charged lipids). Black dots stays for the empty protein, blue for the PS-loaded (incubated with POPS containing LUVs B), and yellow for the PI4P-loaded one (incubated with PI4P containing LUVs B). **C** FCCS temporal cross-correlation curves of Osh6-Atto488 and DiD-labeled LUVs of different composition: POPC + 20 mol% diphytanoylPS (black), POPC + 20 mol% diphytanoylPG (red), POPC + 20 mol% diphytanoylPA (blue). The added Osh6 was either empty (solid lines), or pre-incubated with PS-containing unlabeled LUVs (dotted lines), or with PI4P-containing unlabeled LUVs (dashed lines). Composition of the experiment in 5B, C: $c_{Osh6-Atto488} = 10$ nM, $c_{total\ lipids\ LUV\ A} = 200$ μM, $c_{total\ lipids\ LUV\ B} = 7.5$ μM. All error bars represent the standard error of the mean, n = 10 measurements. The p-values (marked as asterisks) were obtained from the two-sample t-test.

cargo to a membrane containing PI4P and a high level of PS, Osh6 was incubated with LUVs type B, which contained 20 mol% of PS. As observed in our prior experiments (see Fig. 2), Osh6 promptly extracts PS from these LUVs. Subsequently, LUVs type A and SidC-Atto488 were introduced.

Initially, LUVs type A contained 1 mol% of PI4P and no other charged lipids. As depicted in Fig. 7A, B, akin to the findings illustrated in Fig. 4, the presence of PS in LUVs type B halted the extraction of PI4P (Fig. 7A, left panel, blue line).

In the subsequent phase, LUVs type A contained 1 mol% of PI4P along with an additional 20 mol% of PS. When Osh6 was not preincubated with LUVs B, the extraction of PI4P was compromised (Fig. 7A, right panel, black line) due to competition with PS. When LUVs type B comprised solely of pure POPC, the extraction was significantly impeded (Fig. 7A, right panel, red line), albeit not completely inhibited. Most probably, both ligands are extracted at a ratio favoring PS (not visible in this experiment). The additional available space for release drives both cargoes to LUVs B. Eventually, this causes a visible drop in PI4P in LUVs A. Effective extraction of PI4P was only observed when LUVs type B contained 20 mol% of PS (Fig. 7A, right panel, blue line). This indicates that PS-rich LUVs type B not only afford additional space for PI4P delivery but also that Osh6, laden with PS (from LUVs B) compared to the empty one, increases the likelihood of PI4P extraction from membranes abundant in PS.

Taken together, this report underscores a critical mechanistic factor in transport, specifically, that Osh6 is capable of extracting PI4P from the plasma membrane, which is rich in PS, only when it carries the PS cargo and there is space available for PI4P deposition. It also suggests that Osh6, when loaded with PS, can offload the cargo into a plasma membrane-like environment, i.e., a membrane already rich in PS.

Next, we examined whether the PS-loaded Osh6 requires PS in the PI4P-containing membrane to extract it, or whether non-specific charged lipids would be sufficient. We have prepared PI4P (1 mol% PI4P) containing LUVs A with only 1 mol% of PS (instead of 20 mol%) and the rest of PS was replaced by PG (19 mol%) and repeated the experiment. First, Osh6 was not preincubated with LUVs type B. The presence of only 1 mol% of PS already competes with the PI4P extraction as shown on Fig. S3. Upon incubation with LUVs B composed of POPC, the PI4P extraction increases. With PS-loaded Osh6 resulting from the incubation with LUVs B composed of POPC/PS (20 mol%), the PI4P extraction is even more intensified. This

suggests that common charged lipids in the membrane with two cargoes (PI4P/PS) increase the probability of the PI4P uptake for PS-loaded Osh6.

The previous experiments indicate that for the PI4P extraction in the presence of PS, the charge lipids matter. Therefore, we have addressed the kinetics of PI4P extraction from single type of LUVs that, except for 1 mol% of PI4P, contain another charged lipid. Figure 7C depicts extraction rates for various negatively charged headgroups, showing that all of them, except for PS, which is a ligand, accelerate PI4P extraction.

The negatively charged lipids play an opposite role in PS extraction. When LUVs containing 1 mol% are enriched with a charged lipid (PA, PI, PG), the PS extraction rate is reduced as the overall amount of the charged lipids increases (Fig. 7D). When the membrane composition is also enriched with PI4P in an equimolar ratio to PS, the extraction of PS is entirely compromised.

It becomes apparent that these relatively subtle changes in extraction and release rates, along with nuanced variations in Osh6-membrane interaction with different ligands, shift the probability of cargo preference at the moment of its exchange.

### The cargo dependent protein dynamics

To get insights into molecular details of Osh6 motions, we performed all-atom molecular dynamics (MD) simulations of the POPS-loaded and PI4P-loaded Osh6 in solution. For each of the two systems, we conducted three simulations of 500 ns each, resulting in a total of 3 μs of MD data for analysis. To quantify segmental motions of Osh6 with the two cargo lipids, we computed the root-mean-square fluctuations (RMSF) along the Osh6 primary structure (Fig. 8A). We found that the most mobile segment was the C-terminal stretch comprising amino acid residues from Thr420 to Lys435. The N-terminal segment comprising amino acid residues from Ile35 to Gly47 was displaced and quite mobile as well. The most flexible loops were those comprising amino acid residues from Lys206 to Lys211 and from Lys256 to Thr262. These loops and segments were found to be more-or-less equally mobile in each of the Osh6 load states. In contrast, the tip of helix 11, comprising amino acid residues from Glu355 to Pro374, was found to be somewhat more mobile in the POPS-loaded Osh6 than in the PI4P-loaded Osh6, which can be explained by a smaller number of hydrogen bonds between POPS and Osh6 than between PI4P and Osh6 (Fig. 8B). Specifically, we found Glu355 and Arg359 (within

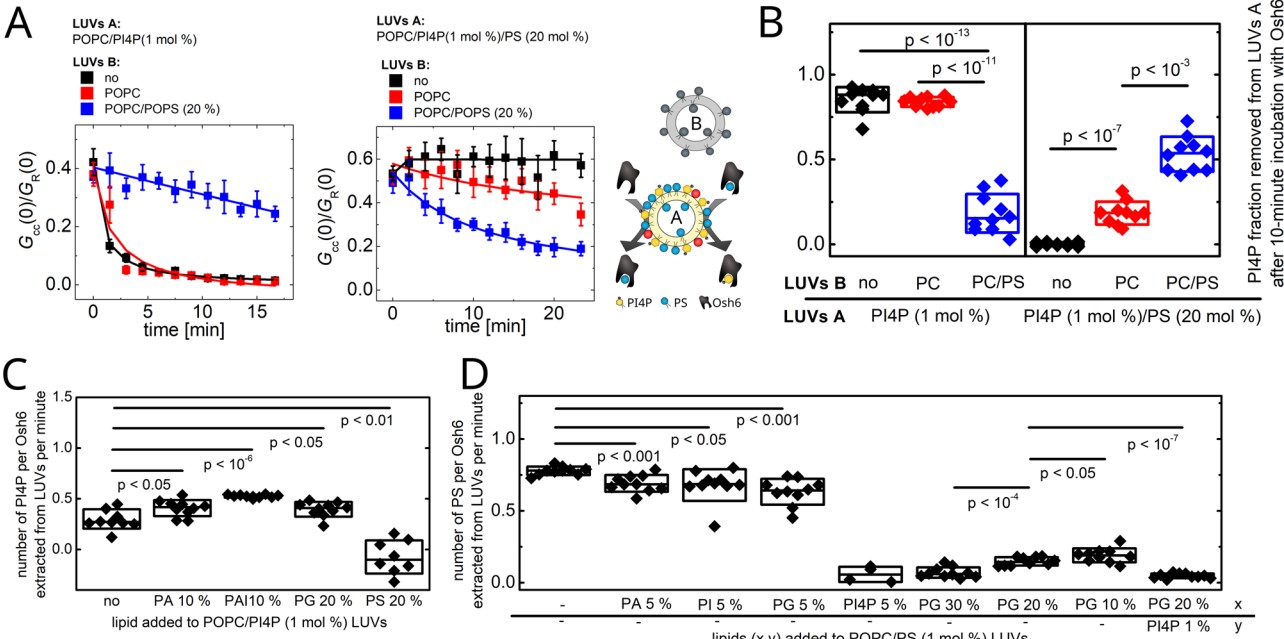

**Fig. 7 | Synchronization of charged lipids and cargo occupancy required for the PI4P extraction from the PM-like membrane. A** Comparison of PI4P extraction by Osh6 from LUVs A that are otherwise neutral (left), and rich in PS (right). Osh6 was preincubated with LUVs B, which (i) supply Osh6 with PS, and (ii) in the final mixture, serve space for PI4P release. LUVs B were composed of POPC/PS (20 mol %, blue curves), POPC (red curves) or were absent (black curves). **B** Extracted fraction of PI4P within first 10 min of the experiment for compositions of LUVs A and LUVs B given in (**A**). **C** Extraction rate of PI4P from the LUVs containing other

charged lipids. **D** Extraction rate of PS from the LUVs containing other charged lipids. Composition of the experiment in 6 A, B: $c_{PI4P}$ = 500 nM, $c_{SidC-Atto488}$ = 100 nM, $c_{total\ lipids\ LUV\ A}$ = 50 μM, $c_{total\ lipids\ LUV\ B}$ = 200 μM; in 6 C: $c_{PI4P}$ = 500 nM, $c_{Osh6}$ = 250 nM, $c_{SidC-Atto488}$ = 100 nM, $c_{total\ lipids\ LUV\ A}$ = 50 μM; in 6D: $c_{PS}$ = 500 nM, $c_{Osh6}$ = 250 nM, $c_{C2lact-CFP}$ = 50 nM, $c_{total\ lipids\ LUV\ A}$ = 50 μM. All error bars represent the standard error of the mean, n = 10 measurements. The p-values were obtained from the two-sample t-test.

helix 11) forming hydrogen bonds with PI4P and not with POPS (Fig. 8C). While the segmental motion of the N-terminal lid was not significantly different for both variants of Osh6, we observed a considerable difference in the number of hydrogen bonds between the lid and the bound lipids (Fig. 8C), particularly involving Arg66 in the case of PI4P-loaded Osh6. This difference may lead to a lower propensity for the lid to open upon contact with the membrane, thus potentially reducing its sensitivity to the negatively charged surface.

As already emphasized by Lipp et al. [16], the N-terminal lid contains the D/E-rich motif that compensates for the basic surface of Osh6 that is below the lid. Lipp therefore considers the Δ69 mutant of Osh6 as an "open" form of Osh6. In order to study how Osh6 interacts with lipid bilayers on the timescale of the order of 100 μs, we performed coarse-grained MD simulations using the Martini 3 model[26]. We simulated four systems: (1) Δ35, the "closed" form in contact with a POPC bilayer, (2) Δ35 in contact with a charged bilayer comprising POPC and POPS in 4:1 molar ratio, (3) Δ69, the "open" form in contact with the POPC bilayer, and (4) Δ69 in contact with the POPC-POPS bilayer. We observed multiple events of binding and unbinding of Osh6 to/from the lipid bilayer in the simulation trajectories (see Figs. S4–S11 in the supplementary material). For each of the four systems we thus monitored when and which of the amino acid residues were in contact with the lipid bilayer (see Figs. S4–S11 in the supporting material). Figure 8D, E show the probability of Δ35 making contacts with the POPC and POPC-POPS bilayers, respectively. The amino acid residues that are found to contact the bilayer are Phe229, Tyr258, Val259, Phe260, Pro297 and Arg297 (Fig. 8F). They are localized within three flexible loops. Figure 8G, H show the probability of Δ69 contacts with the POPC and POPC-POPS bilayers, respectively. The membrane contact site is formed by three segments: from Arg184 to Ser189, from Arg225 to Arg233, and from Lys256 to Tyr263 (Fig. 8I). Thus, the orientations of Δ35 and Δ69 at the lipid bilayer are found to be somewhat different. The data shown in Fig. 8D–I demonstrate together that Δ69 associates with lipid bilayers more strongly than

Δ35, and that the electrostatic charge on the bilayer enhances the association of both Δ69 and Δ35.

Little is still known about the exact mechanism of lipid extraction and release, as it requires microsecond-long all-atomistic simulations to visualize the interaction between the membrane and the protein. The coarse-grained model used here does not allow for the observation of larger segmental mobility of Osh6. However, some work has already been done with the ceramide-transporting protein CERT[27].

## Discussion

In this study, we have conducted a comprehensive biophysical analysis of the lipid transfer protein Osh6 and its role in lipid transportation. While we acknowledge that other biological factors, such as regulatory proteins, energy-loaded pools of PI4P, and membrane tethers, also play crucial roles in this process, the fundamental biophysical interplay between the transporter's structure, its cargo, and the characteristics of the involved membranes serve as the underlying mechanistic platform of this process.

Most of the studies focus on the Osh6 transporter and its cargo molecules of various head groups and fatty acyl chains[15,23,28] without addressing the milieu accommodating the cargo—the membranes. In the case of Osh4, a sterol transporter, the saturated fatty acyl chains in the sterol accepting membrane help stabilize the sterol gradient that needs to be established between the membrane of trans-Golgi and ER[12]. Additionally, the interaction between the distal membrane binding site of Osh4 and PIP$_2$ has been identified to play a part in the sterol transport[10]. Thus, the molecular view of the transport dynamics does not only involve the protein/ligand interactions but also the protein/membrane, as well as the cargo lipid–ligand/membrane interactions. Moreover, in addition to the thermodynamic affinities between the key players, it turns out that the kinetics of the individual transport steps also come into play.

To ensure the biological relevance of our investigation, we pose questions that are significant for PS transport in cell-like membranes. One

**Fig. 8 | (A-C) Results of the all-atom MD simulations of the POPS-loaded (red) and PI4P loaded (blue) Osh6 in solution.** The error bars indicate standard deviation obtained from three independent trajectories. **A** RMSF values along the Osh6 primary structure. **B** Distribution of the number of hydrogen bonds between Osh6 and the cargo lipid. **C** Occupancy of hydrogen bonds between the cargo lipid and helix 11 as well as between the cargo lipid and the lid. **D–I** Results of the coarse-grained MD simulations. Probability of contact of Δ35 with the POPC (**D**) and POPC-POPS (**E**) bilayer. Results obtained from two trajectories. **F** Cartoon illustrating how Δ35 associates with a lipid bilayer. Δ35 is colored red-to-blue from the N- to C-terminus. The amino acid residues that get inserted into the lipid bilayer are shown in the van der Waals representation. (**G, H**) Analogous to (**D**) and (**E**) but for Δ69. **I** Analogous to (**F**) but for Δ69.

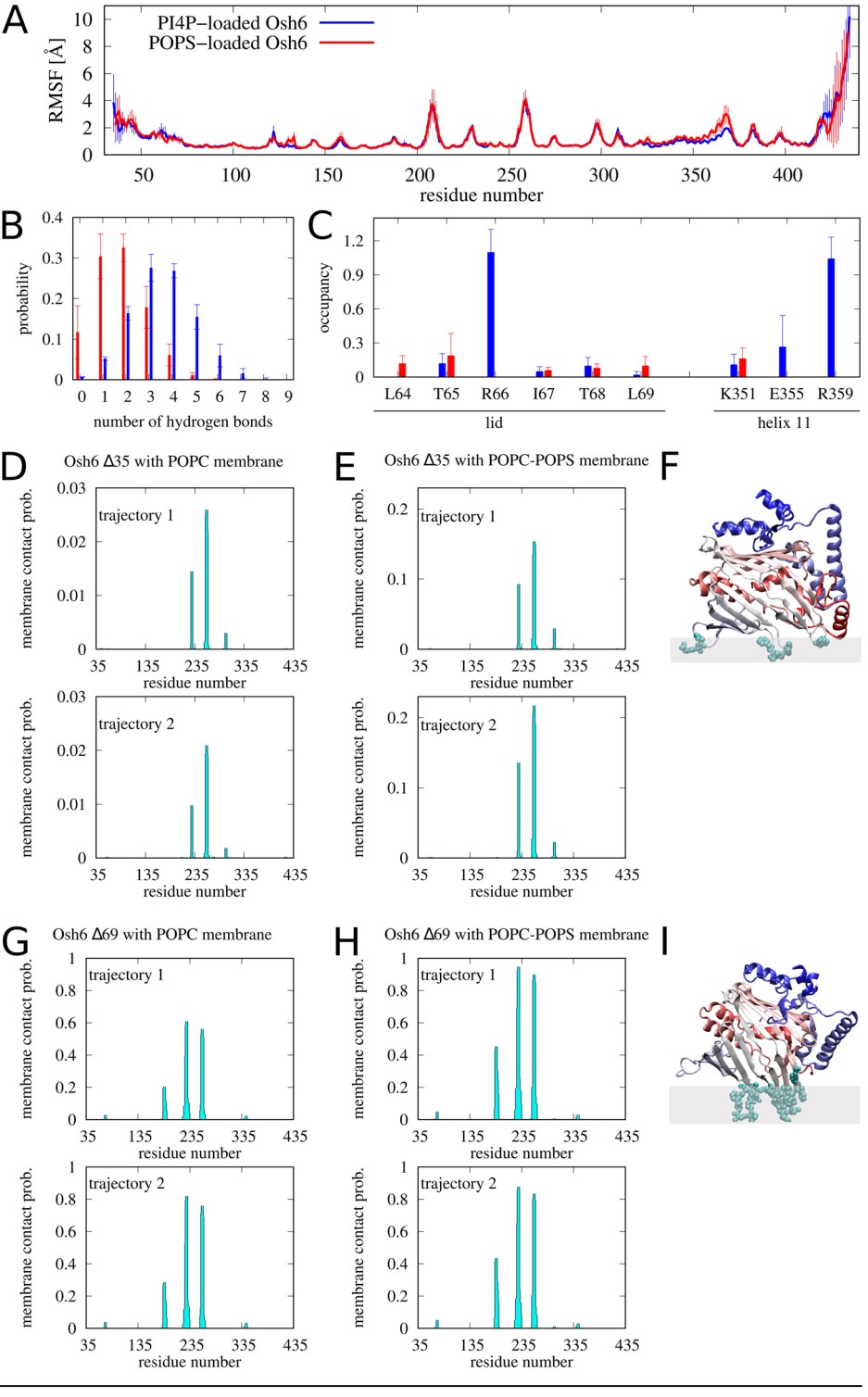

intriguing question is related to the fact that PM contains more than 20 mol % of PS, predominantly in its inner leaflet, compared to the low level of PS in ER[1]. Therefore, PS transport needs to occur against this large concentration gradient. Even though PI4P is a stronger binder of Osh6[7], statistically, the exchange of PS delivered by Osh6 from ER to the PM surface would favor the far more abundant molecule—PS. In other words, no enrichment of PM by PS would occur. Additionally, our findings demonstrate that the extraction of PS from the neutral membrane is significantly faster than that of PI4P. It should be noted that while PS can be spontaneously and independently delivered along the gradient, the experiments examining the

extraction of PI4P from PS-rich LUVs indirectly, yet convincingly, suggest that PS transport against the gradient is enabled by the negative charge of PS and by the presence of PI4P for exchange in the PS-accepting membrane. The membrane rich in charged lipids promotes the release of PS, accelerates the uptake of PI4P, and inhibits the non-effective re-extraction of PS. From the perspective of the cargo, PS-loaded Osh6 is poised to be unloaded at the PM surface.

The reason that the negative charge promotes the counter-gradient transport of PS is likely not only because it facilitates lid opening, but also, from a thermodynamic perspective, due to favorable interactions between

the delivered PS molecule and other charged lipids, making PS reuptake energetically unfavorable. This is supported by the kinetic decays in Fig. 3D–F, which show that over time, the largest number of PS is transported to the charged membranes. Additionally, Fig. 7D shows that the more charged lipids the membrane contains, the less PS can be exported from it. Previous studies[29] have shown that in mixed POPC/POPS membranes, more POPS-POPS contacts are made than POPS-POPC contacts, suggesting that charged lipids attract each other, thereby stabilizing them within the membrane.

The other question to be answered concerns PI4P offloading. Our experiments show that, in contrast to PS, PI4P disembarkation is not enhanced by charged lipids other than cargo. Generally, in line with the higher affinity of Osh6 for PI4P compared to PS, PI4P-loaded Osh6 is significantly more retentive. The release is enhanced in soft, low-charged membranes that are endowed by phosphatase to shift the equilibrium[13,15], such as Sac1 in the ER membrane.

Examination of the molecular dynamics of POPS- and PI4P-loaded Osh6 in solution through all-atom simulations revealed a greater mobility of the N- and C-terminal segments. As suggested by previous research[16], an N-terminal lid containing the D/E-rich motifs shields a basic patch of the protein surface. This indicates that the increase in lipid release by negatively charged membranes reflects the involvement of this basic patch in a partially open Osh6 structure. One key finding from the all-atom simulations that may help elucidate the difference in charge-dependent behavior between PS- and PI4P-loaded Osh6 is the disparity in mobility of helix 11. Notably, this helix forms more hydrogen bonds with PI4P than with POPS and encompasses one of the key residues (Arg359) in the coordination shell of PI4P[7]. Additionally, the N-terminal lid may contribute to the higher sensitivity of PS-loaded Osh6 to the negatively charged surface by forming fewer hydrogen bonds with the lipid headgroup compared to PI4P-loaded Osh6.

To complement the all-atom simulations of Osh6 in solution, Martini coarse-grained simulations of the protein were performed to examine its interactions with charged and neutral membranes. The lidless Δ69 mutant represented the "open" form of the protein, and the Δ35 mutant was considered the "closed" one. The "open" form showed a significantly stronger association with the membrane, especially with the negatively charged one. This finding additionally confirms that our ~150 μs Martini simulations provide predictions consistent with those obtained from the ~500 ns all-atom simulations of Lipp et al.[16].

In accordance with the work of Lipp et al., our findings confirm that the membrane interaction of Osh6 is reduced when the membrane contains the cargo, and the reduction is closely related to the status of the N-terminal lid. In our work, we furthermore claim that the type of cargo governs the behavior of the protein's action at the specific moment of the PS transport cycle, i.e., PS- and PI4P-loaded Osh6 is determined to offload the cargo in PM and ER, respectively.

Our study provides a detailed insight into the biophysical orchestration of lipid transport by the lipid transfer protein Osh6. We demonstrate that the cargo itself induces changes in the protein dynamics and affects its propensity to open its lid. This influences the transporter's response towards the appropriate target membrane. While carrying PS loosens the lid and exposes the transporter's basic surface to release the cargo into highly charged membranes (such as the PM), carrying PI4P tightens the lid, necessitating a low-charged, fluid membrane (such as the ER) to accommodate the PI4P unloading.

## Methods

### Protein expression and purification

The genes encoding Osh6, SidC, and C2$_{Lact}$-CFP were cloned into a modified pHIS2 vector containing an N-terminal His6x-tag followed by a tobacco etch virus (TEV) cleavage site. Mutations to produce cysteine-less protein (C62S/C162S/C389S) and a single cysteine mutant (I241C) were generated using site-directed mutagenesis. The proteins were expressed in E. coli BL21 Star cells using our standard protocols[30,31]. The cells were

harvested by centrifugation, resuspended in a lysis buffer (50 mM Tris, pH = 8, 300 mM NaCl, 20 mM imidazole, 3 mM β-mercaptoethanol, and 10% glycerol), and lysed by sonication. The lysates were cleared by centrifugation and incubated with nickel-charged affinity resin (Machery-Nagel). The proteins were eluted with an elution buffer (lysis buffer supplemented with 300 mM imidazole), and the His6x-tag was cleaved off by TEV protease.

Subsequently, the proteins were purified by size exclusion chromatography on a HiLoad 16/600 Superdex 75 pg column (Cytiva) in 20 mM Tris, pH 7.4, 300 mM NaCl, 3 mM β-mercaptoethanol, and 10% glycerol. Osh6 was further purified by ion-exchange chromatography on a HiTrap SPHP column (Cytiva).

For Atto488 labeling, the single cysteine mutant of Osh6 and SidC were transferred to PBS and mixed with a 3-fold molar excess of Atto488 maleimide. After overnight incubation at 4 °C, the unbound dye was removed by size exclusion chromatography on Superdex 200 pg (Cytiva) in 20 mM Tris, pH 7.4, 300 mM NaCl, 3 mM β-mercaptoethanol, and 10% glycerol.

### Lipids and other chemicals

All lipids except for di16:0 PI4P were purchased from Avanti Polar Lipids (Alabaster, AL) and were used as obtained. Di16:0 PI4P was obtained from Echelon Biosciences (Salt Lake City, UT) and also used as obtained. Atto488-labeled DOPE was obtained from ATTO-TEC (Siegen, Germany), and the lipid tracer DiD and other basic chemicals were purchased from Sigma-Aldrich (St. Louis, MO).

### LUV formation

Lipids in organic solvents were mixed in the desired ratio so that the final lipid concentration in LUVs was 1 mM. Organic solvents were evaporated in a stream of nitrogen and kept in the vacuum chamber for at least one hour. Later, the lipid films were resuspended in the LUV buffer (40 mM imidazole (pH = 7.4), 150 mM NaCl, 3 mM beta-mercaptoethanol, 1 mM EDTA), and 50 nm LUVs in diameter were prepared in the extruder through the membrane of an appropriate pore size[32].

### Kinetic assays

All the kinetics was acquired as a short 200-s measurement prior the transporter addition and a longer (5 to 20 min) measurement upon its addition. The Osh6 concentration was 250 nM. The concentration of the biosensors C2$_{Lact}$-CFP and SidC-Atto488 in the total volume of 200 μL was 50 and 100 nM, respectively.

### PS extraction assays

10 μl of donor LUVs containing POPC (94 mol%), diphytanoyl-PG (5 mol %), POPS (1 mol%) and DiD were mixed with C2$_{Lact}$-CFP, the LUV buffer and 0 μl, or 40 μl of unlabeled LUVs composed of POPC and various contents of other lipid species, as indicated in related figures (Figs. 2C, 7D).

### PS transport assays

10 μl of donor LUVs containing POPC (91 mol%), diphytanoyl-PG (5 mol %), POPS (4 mol%) and DiD were mixed with C2$_{Lact}$-CFP, the LUV buffer and 0 μl or 40 μl of unlabeled LUVs of various lipid compositions as indicated in Fig. 3D–F.

### PI4P extraction assays

10 μl of donor LUVs containing POPC (99 mol %), PI4P (1 mol %) and DiD were mixed with SidC-Atto488, the LUV buffer, and 0 μl or 40 μl of unlabeled LUVs composed of POPC and various amounts of other lipids, as indicated in Fig. 4C.

In the experiments that required loading of Osh6 with PS (Fig. 7B), Osh6 was pre-incubated with 20 μl of unlabeled LUVs composed of POPC/POPS (80/20 mol%) for 10 min. Next, 20 μl of DiD labeled LUVs composed of POPC/PI4P (99/1 mol%) or POPC/PI4P/POPS (79/1/20 mol%) was mixed with the LUV buffer and SidC-Atto488. Subsequently, a short 200-second measurement was started. The pre-incubated sample of Osh6 and

POPC/POPS LUVs were then added, and immediately after that, another 15-minute FCCS measurement was started. The concentration of Osh6 in the final 200 μl sample was 250 nM, and the concentration of SidC-Atto488 was 100 nM.

## PI4P transport assays

10 μl of donor LUVs containing POPC (97 mol%), PI4P (3 mol%) and DiD were mixed with SidC-Atto488, the LUV buffer, and 0 μl or 40 μl of unlabeled LUVs composed of lipid mixtures, as indicated in Fig. 5B–D and Fig. 7C.

## Osh6 membrane binding

Osh6-Atto488 was incubated with unlabeled LUVs composed of DOPC/POPS (80/20) or DOPC/PI4P (95/5) for 10 min. Subsequently, DiD-labeled LUVs of different compositions specified in Fig. 6B were added so that in the 200 μl total volume, the final concentration of unlabeled and labeled LUVs was 7.5 μM and 200 μM, respectively. The mixture was incubated for 10 min, and a 200-s FCCS measurement was carried out. In the case of empty Osh6, Osh6-Atto488 was added directly to DiD-labeled LUVs. The final concentration of Osh6-Atto488 in all samples was 10 nM.

The labeling of LUVs with DiD was done in the DiD/lipid = 1/10,000 ratio. All the FCCS experiments were acquired at least in three independent experiments to ensure for reproducibility.

## Microscopy

The FCCS experiments were carried out on the Leica SP8 confocal microscope (Leica, Mannheim, Germany) equipped with a high numerical aperture water objective (63x, N.A. = 1.2), a battery of synchronizable pulsed lasers, and sensitive hybrid HyD detectors. In our experiments, we used the 640 nm line of the white light laser (Coherent, Inc., Santa Clara, CA) for DiD excitation and the 440 nm and 470 nm diode laser heads (Picoquant, Berlin, Germany) for CFP and Atto488 excitation, respectively. The pair of lasers (440/640 and 470/640) was pulsing alternatively at the pulsed interleaved excitation (PIE) mode at an overall repetition frequency of 40 and 20 MHz, respectively. The PIE mode was used to apply temporal filtering of photon arrival times in addition to the spectral information to omit bleed-through. The data were correlated and evaluated by home-written scripts in Matlab (Mathworks, Natick, MA).

## Estimation of lipid extraction and lipid transport rates

In the case of extraction, the initial extraction rate $v_{ext}$ was evaluated according to the following formula:

$$v_{ext} = \frac{c_{lipid(acc.)}}{c_{protein}} \times \frac{\left(G_{cc}(0)/G_R(0)\right)_{t=0} - \left(G_{cc}(0)/G_R(0)\right)_t}{\left(G_{cc}(0)/G_R(0)\right)_{t=0} \times t}, \quad (2)$$

where $c_{lipid(acc.)}$ and $c_{protein}$ represent the concentration of the accessible cargo lipid in the donor LUVs and of the transfer protein, respectively. $(G_{cc}(0)/G_R(0))_{t=0}$ and $(G_{cc}(0)/G_R(0))_t$ denote the values of the FCCS readout parameter before the transporter is added and at time $t$ after the addition, respectively. For the fast extraction of PS, $t$ was set to 1 min (Figs. 2E and 7D), which is the resolution limit of our experiment. For the slower extraction of PI4P, $t$ was set to 3 min (Figs. 4E and 7C) to better capture differences in kinetic decay.

In the case of the transport rate, we have evaluated the average transport rate $v_{transport}$ according to the following formula:

$$v_{transport} = \frac{\left(c_{lipid(acc.)} - c_{protein}\right)}{c_{protein}} \times \frac{\left(G_{cc}(0)/G_R(0)\right)_{t_1} - \left(G_{cc}(0)/G_R(0)\right)_{t_2}}{\left(G_{cc}(0)/G_R(0)\right)_{t_1} \times (t_2 - t_1)},$$

$$(3)$$

where $t_1$ is set to the time when the majority of the protein is occupied by the cargo, i.e., when the extraction process has finished. This occurs at 1 min and 6 min for PS and PI4P transport, respectively. $t_2$ was selected to be long

enough to capture differences among the individual decay processes: 10 min for PS (Fig. 3G–I) and 16 min for PI4P (Fig. 5G). In the formula, the accessible concentration of the lipid in the donor LUVs is adjusted to account for the fact that, once the extraction is complete, the available lipid for transport is reduced by the lipid already inside the transporter.

In Fig. 7B, the overall drop of PI4P ($\Delta_{PI4P}$) in LUVs A during the first 10 min after protein addition was monitored. The drop was quantified regardless of whether it occurred through extraction or transport. We have thus modified Eq. 2 as follows:

$$\Delta_{PI4P} = \frac{c_{lipid(acc.)}}{c_{protein}} \times \frac{\left(G_{cc}(0)/G_R(0)\right)_{t=0} - \left(G_{cc}(0)/G_R(0)\right)_{t=10min}}{\left(G_{cc}(0)/G_R(0)\right)_{t=0}}. \quad (4)$$

## All-atom MD simulations

The atomic coordinates of the POPS-loaded and PI4P-loaded Osh6 were taken from the crystal structures deposited in the Protein Data Bank (PDB) with the entry codes of 4B2Z and 4PH7, respectively[7,8]. Systems for MD simulations were prepared using the input generator on the CHARMM-GUI website[33,34]. Each of the protein structures was solvated in a cubic box with the side length of 9 nm. Sodium and chloride ions were added to neutralize the systems and to reach a physiological ion concentration of 150 mM. The initial systems for MD simulations were energy-minimized using a conjugate gradient method and then equilibrated in a standard procedure using input files provided by the CHARMM-GUI input generator.

The MD simulations were performed using NAMD 2.14 with CHARMM36 force field and the TIP3P model for water molecules[35–38]. Temperature was kept at 303 K through a Langevin thermostat with a damping coefficient of 1/ps. Pressure was maintained at 1 atm using the Langevin piston Nose-Hoover method with a damping timescale of 25 fs and an oscillation period of 50 fs. Short-range nonbonded interactions were cutoff smoothly between 1 and 1.2 nm. Long-range electrostatic interactions were computed using the particle mesh Ewald method with a grid spacing of 0.1 nm. Simulations were performed with an integration time step of 2 fs.

For each of the simulation systems (i.e., the POPS- and PI4P-loaded Osh6) we performed three production runs of 500 ns each, amounting to 3 μs of MD data for analysis. The simulation trajectories were visualized and analyzed using VMD[39].

## Coarse-grained MD simulations

Two variants of Osh6 in contact with lipid membranes were simulated using the Martini 3 model: Δ35 (comprising amino acid residues with numbers from 36 to 434) and Δ69 (comprising amino acid residues with numbers from 70 to 434). The atomic coordinates of Δ35 and Δ69 were taken from the crystal structures deposited in the PDB with the entry codes of 4PH7 and 4B2Z, respectively[7,8].

Systems for coarse-grained MD simulations were set up in the following way using the Martini maker on the CHARMM-GUI input generator website[33,40]: Two bilayer segments with equal lateral dimensions of 12 nm by 12 nm were formed independently. In one case, the bilayer was composed of 352 POPC lipids and 88 POPS lipids (i.e., with 4:1 molar ratio). In the other case, the bilayer was composed of 422 POPC lipids. Then each of the two Osh6 structures was placed in a random orientation about 4 nm above each of the lipid bilayers, which produced four simulation systems, i.e., Δ35 with the POPC-POPS bilayer, Δ35 with the POPC bilayer, Δ69 with the POPC-POPS bilayer, and Δ69 with the POPC bilayer. Each of these systems was placed in a cuboid box and solvated. Sodium and chloride ions were added to neutralize the systems and to reach a physiological ion concentration of 150 mM. The simulation systems were coarse-grained within the framework of the Martini 3 model with an elastic network (ELNEDIN) applied to Osh6 beads[26,41]. The initial systems for MD simulations were energy-minimized using a conjugate gradient method, and then equilibrated in a standard procedure using input files generated by the Martini maker in the CHARMM-GUI input generator.

The coarse-grained MD simulations were performed using Gromacs 2020.2 and the Martini 3.0 force field[26,42,43]. The temperature and pressure were kept constant at $T = 303$ K and $p = 1$ bar, respectively, using the velocity-rescaling thermostat and the Parrinello-Rahman barostat[44,45]. Nonbonded interactions were treated with the Verlet cutoff scheme. The cutoff for van der Waals interactions was set to 1.1 nm. Coulomb interactions were treated using the reaction-field method with a cutoff of 1.1 nm and dielectric constant of 15. The integration time step was set to 20 fs. For each of the four systems we performed two simulation runs of 70 μs each, yielding a total of 560 μs of MD data for further analysis. Frames were saved every 1 ns. The simulation trajectories were post-processed with MDVWhole to treat the periodic boundary conditions, and visualized using VMD[39].

## Statistics and reproducibility

All kinetic curves, corresponding rates, and cross-correlation amplitudes (Figs. 2–7) were obtained from ten experimental points measured with the same stock of LUVs. This approach was chosen because independent experiments with LUVs, a self-assembly system, and FCCS, which has high sensitivity to the number of fluorophores per particle, showed lower reproducibility in the absolute values of the read-out parameter, $G_{cc}$. Data points were excluded if their deviation from the mean exceeded 1.5 times the standard deviation. The remaining data were used to calculate the mean, its error, and p-values via a t-test. To ensure reproducibility, all trends presented in the manuscript were measured in at least three independent experiments.

## Data availability

Source data underlying graphs can be found in Supplementary data. Additional supporting data of any kind are available from the corresponding authors upon request.

## Code availability

We did not use any special code or algorithm central to the manuscript. All MATLAB scripts we used are fully available through the LAS X software on the Falcon-equipped SP8 microscope.

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

## Acknowledgements
The research was supported jointly by the Czech Science Foundation, grant number 21-27735 K (to J.H. and E.B.), and by the National Science Centre, Poland, grant number 2020/02/Y/NZ1/00020 (to B.R.), within the international CEUS-UNISONO program. The molecular dynamics simulations were performed using the supercomputer resources at the Centre of Informatics–Tricity Academic Supercomputer and networK (CI TASK) in Gdansk, Poland. We also acknowledge the Academy of Sciences of the Czech Republic (RVO: 61388963). E.B. would like to thank the project New Technologies for Translational Research in Pharmaceutical Sciences/NET-PHARM, project ID CZ.02.01.01/00/22_008/0004607, co-funded by the European Union.

## Author contributions
A.B. and J.H. designed the experiments. A.B. conducted the FCCS measurements, while JH evaluated the data. A.E. prepared, purified, and labeled the proteins. B.R. carried out the simulations. E.B. supervised the project. J.H. wrote the manuscript, and all contributors participated in the data interpretation and manuscript preparation.

## Competing interests
The authors declare no competing interests.
