## [Transparent Peer Review file · Communications Biology]

Coordination of transporter, cargo, and membrane properties during non-vesicular lipid transport.

Corresponding Author: Dr Jana Humpolickova

Version 0:

Reviewer comments:

Reviewer #1

(Remarks to the Author)

Comments to COMMSBIO-24-3770-T

Biophysical scrutinizing “extraction” and “release” steps separately in a lipid transfer protein (LTP)-mediated process should provide fundamental insight into comprehending the LTP-mediated inter-organelle trafficking of lipids in cells. The manuscript by Ballecova et al. (the authors) attempted dissection of Osh6-dependent non-vesicular transport of PS and PI4P, employing the FCCS technique and molecular dynamic simulation. Although this attempt is challenging, several critical concerns about the interpretations of the data in the manuscript must be amended. Specific comments are listed below.

Major comments

1) L101-L124: Biosensor and lipid-transfer protein (Osh6 in this study) recognize the same cargo molecule (PS in this study), meaning that Osh6 acts as a competitor for the biosensor in binding to the lipid ligand. Considering that this competitive action occurs on LUVs, Osh6 is expected to detach the biosensor from LUVs even without the “extraction” of PS. However, the current manuscript neglects the competitive binding effects on interpreting the data. Although the authors state that “the extraction occurs instantaneously” (e.g., L117 and L139), the data may be alternatively interpreted that “Osh6 binding to PS on LUVs occurs instantaneously”. The authors should eliminate this possibility. To address the criticism, this reviewer suggests testing what will happen when the “unextractable” diphytanoyl PS is used instead of extractable PS in donor LUVs A in figure 1. If the addition of Osh6 causes the release of the biosensor labelled-C2-Lact even from the diphytanoyl PS-containing LUVs, such data means detaching of the biosensor occurs without extracting the cargo. In this case, the authors should reconsider the logical basis of the FCCS part of this study.

2) Regarding the above concerns, testing what will happen when the unlabelled C2-Lact is added instead of Osh6 may also be interesting. This reviewer predicts that unlabelled-C2-Lact acts as a PS-binding competitor and causes an instantaneous detachment of labelled-C2-Lact. If this is the case, the detaching of the biosensor does not monitor the release of the cargo from the donor (because the unlabelled-C2-Lact can act as a binding competitor but not as a PS-extraction module). In contrast, if the genuine binding competitor “unlabelled-C2-Lact” exhibits an output pattern quite different from the case with Osh6, it would be evidence against the concern. In addition, the experiments with unlabelled-C2-Lact may be a good control to show that “PS deposition to acceptor LUV” can occur by the PS-transfer protein Osh6 but not by the unlabelled-C2-Lact. A similar concern occurs in the PI4P extraction experiments (Fig. 3).

3) Figs. 2 and 3: How did the authors determine the “numbers of removed PS (or PI4P) per Osh6 to LUVs per minute”? Compared with the data of Fig. 2D-F, the data of Fig. 2G-I seemingly represent the plateau levels (e.g., 15-20-min point), not initial rates (e.g., 1-2-min point). Consistent with this guess, the authors noted that “the kinetics of PS extraction cannot be captured within the temporal resolution of our experiment” (L256-257), meaning it is infeasible to determine the initial rate of PS removal. Thus, the statement (L256-257) would be contradicted if “200-s points” were used for PS extraction kinetics (L573-576). Overall, the authors should clearly describe how the “numbers of transported (or extracted) lipid per Osh6 to LUVs per minute” were determined. Then, the authors should clarify whether they describe thermodynamically equilibrated states (i.e., plateau levels) or non-equilibrated transient states (i.e., kinetic rates) in the text. Due to the ambiguity of the term used in figures, it is difficult for me to assess the thermodynamic soundness of the conclusions from the part of FCCS analysis.

4) L196-225 and Fig. 2: This reviewer has not been convinced by the main conclusion that lipid compositions of acceptor LUVs significantly affect the release of PS to the acceptor membrane. As the cell-free assay system used in this study did not input any biological energy, this assay should attain thermodynamically equilibrium states at the steady state. Thus, the

data shown in Fig 2 (and also in Fig. 4 for PI4P) may be simply due to the dependence of thermodynamically equilibrated mol% of PS (and PI4P) in LUVs on co-existing lipids and membrane fluidity, not primarily on the lipid releasing speed of Osh6. The authors should address this point. Energetic and/or thermodynamic aspects of LTP-mediated lipid transfer have been described in previous reviews (J Lipid Res, 2018, 59, 1341; J Cell Biol, 2021, 220, e202012058).

Minor comments:

5) The results show that PI4P in donor LUVs inhibits PS extraction (Fig. 1D, E). I agree with the explanation that PI4P serves as a binding competitor of PS for Osh6. In contrast, it is enigmatic that PI4P in acceptor LUVs inhibits PS release from Osh6 (Fig. 2D, G) because one can imagine that PI4P in the acceptor may accelerate the release of PS from Osh6 by exchanging the captured lipid ligand. Please explain this paradox briefly.

6) L502-503 "PS transport against the gradient is enabled by the negative charge of PS and by the presence of PI4P for exchange in the PS-accepting membrane". I understand the importance of PI4P for PS transport against the gradient from the PS/PI4P-exchanging model. Please add a brief explanation from energetic or thermodynamic aspects of why the negative charge of PS is also important.

7) L42-43: I disagree with the introductory statements, which use an old review article as only one reference. Recent studies have shown that LTP-mediated non-vesicular pathways, rather than vesicular pathways for protein sorting, are predominant for inter-organelle sorting of lipids (e.g., reviewed in J Lipid Res, 2018, 59, 1341; Nat Rev Mol Cell Biol, 2019, 20, 85; J Cell Biol, 2021, 220, e202012058; Annu Rev Cell Mol Dev 2023, 39, 409, and references therein).

8) Regarding the lipid composition dependence on the cargo release into acceptor membranes, it would also be meaningful to touch on the possibility that an acyl chain of a specific lipid type sweeps the cargo out of the hydrophobic cavity of Osh6, as recently shown by an MD study on the CERT START domain (J Phys Chem B 2024: 128, 6338).

9) L48: The term "phosphatidylinositol 4-phosphate (PI4P)" should be "phosphatidylinositol 4-monophosphate (PI4P)" as the authors correctly use the term "phosphatidylinositol-4,5-bisphosphate (PIP2)" in L60.

Reviewer #2

(Remarks to the Author)

This study explores lipid transport, focusing on both the interactions between cargo lipids and proteins and the interactions between the entire membrane and proteins. Using an in vitro approach, the research reconstructs the transport process with a simplified large unilamellar vesicle system. This setup allows for an examination of how certain membrane characteristics, such as the presence of charged lipids and membrane fluidity, influence Osh6-mediated lipid transport. The findings suggest that the cargo itself induces changes in protein dynamics, thereby affecting the transporter's interaction with the target membrane.

The experimental system used in this study provides valuable insights into the biophysical properties of lipid transport and is of significant scientific value. However, the study lacks experiments that address biological relevance. Moreover, Osh6 is a yeast protein; extending this research to homologous proteins in mammals and humans would substantially increase its impact.

Reviewer #3

(Remarks to the Author)

Lipid transfer proteins play wide and important roles in lipid homeostasis. Yet their molecular mechanisms are not well understood. In this manuscript, Ballekova et al. investigated the mechanism of Osh6-mediated lipid transfer. Osh6 is a representative of shuttle lipid transfer proteins and its homologs such as Osh4 have been widely studied. The authors used new fluorescence correlation spectroscopy to detect lipid transfer between membranes and Osh6 and between different membranes. The results are useful for understanding lipid transfer proteins in general. However, the manuscript should undergo a major revision to address the following concerns and comments, before its publication in Communications Biology:

1. It is not immediately clear what new insights into the molecular mechanisms of lipid transfer are brought forward by this study. The authors emphasized the role of membrane properties in lipid transfer, such as non-cognate lipids and membrane fluidity. However, this feature is not new and has been well-demonstrated by many groups, including Drin and coworkers (e.g., Moser von Filseck, J. et al., Nat. Commun. 6, 6671, 2015). The authors seemed to suggest that strong binding between Osh6 and membranes facilitates lipid transfer. However, lipid transfer involves Osh6 unbinding from one membrane and binding to another membrane. Thus, strong membrane binding appears to hinder fast lipid transfer. Consistent with this view, many lipid binding domains in shuttle lipid transfer proteins minimally bind membranes.

2. Osh5 counter-transport both PS and PI4P and the concentration gradients of both PS and PI4P affect lipid transfer. The authors are encouraged to interpret their data more quantitatively based on the latest model of shuttle lipid transfer (Zhang, Y. et al. Contact, 5, 2022). Based on the lipid exchange model, the presence of a cognate lipid facilitates the exchange and transfer of another cognate lipid. Is the finding in this work consistent with this model?

3. The manuscript should be revised to enhance readership. It is difficult to follow experimental descriptions. In the lipid extract assay, both free POPS and POPS-labeled DiD (line 578) were included in the liposome (Fig. 1A). Does Osh6 bind the dye-labeled POPS?

4. Is C2-CFP a specific sensor to PS? Note that many C2 domains broadly bind negatively charged lipids such as PI4P and PIP2.

Version 1:

Reviewer comments:

Reviewer #1

(Remarks to the Author)

The authors appropriately remised the manuscript. Although it almost fully satisfied me, I found two minor flaws in the revised manuscript, as listed below. They should better be amended.

1) By the new Fig. S2, my most pivotal concern on the previous manuscript was removed. I have a minor question on the figure. The value of $G_{cc}(0)/G_R(0)$ at the time zero point in Fig. 1D is about 0.08, while that in Fig. S2 is about 0.25. Why are these two values so different? Please explain the cause of this difference in the legend in Fig S2.

2) "Additionally, during PS transport to PI4P-containing LUVs, the replacement of PS by PI4P halts the cyclic process of PS extraction and release, as the transporter becomes occupied by a stronger ligand and is unavailable for another cycle." Although I agree with this explanation, I suggest putting an additional sentence, for example, "Notably, after releasing PI4P from the lipid transfer domain to acceptor organelle, metabolic conversion of PI4P to PI occurs to continue the cycle within cells" (refs).

Reviewer #3

(Remarks to the Author)

The revision has addressed my concerns. I support its publication.

Reviewer I

Major comments

1) L101-L124: Biosensor and lipid-transfer protein (Osh6 in this study) recognize the same cargo molecule (PS in this study), meaning that Osh6 acts as a competitor for the biosensor in binding to the lipid ligand. Considering that this competitive action occurs on LUVs, Osh6 is expected to detach the biosensor from LUVs even without the “extraction” of PS. However, the current manuscript neglects the competitive binding effects on interpreting the data. Although the authors state that “the extraction occurs instantaneously” (e.g., L117 and L139), the data may be alternatively interpreted that “Osh6 binding to PS on LUVs occurs instantaneously”. The authors should eliminate this possibility. To address the criticism, this reviewer suggests testing what will happen when the “unextractable” diphytanoyl PS is used instead of extractable PS in donor LUVs A in figure 1. If the addition of Osh6 causes the release of the biosensor labelled-C2-Lact even from the diphytanoyl PS-containing LUVs, such data means detaching of the biosensor occurs without extracting the cargo. In this case, the authors should reconsider the logical basis of the FCCS part of this study.

We would like to thank the reviewer for this comment. Indeed, we had neglected to include this interpretation in the manuscript. To ensure that extraction, rather than competition as the reviewer suggests, is responsible, we performed the suggested experiment. As shown in the SI, even with a fourfold excess of Osh6, no drop in $G_{cc}(0)$ occurs, indicating that the binding of the biosensor to the PS head group is apparently stronger than the binding of Osh6.

We have added to the main text:

Additionally, to ensure that the decrease in $G_{cc}(0)$ is not solely due to the competition between C2_{lact}-CFP and Osh6 for cargo binding, we examined the extraction of the non-extractable lipid diphytanoylPS (phPS). As shown in Fig. S2, no change in $G_{cc}(0)$ occurs upon the addition of Osh6, even at a fourfold excess, suggesting that the observed decrease in $G_{cc}(0)$ in Fig. 1D is indeed caused by the extraction of PS, rather than the displacement of the biosensor by Osh6.

2) Regarding the above concerns, testing what will happen when the unlabelled C2-Lact is added instead of Osh6 may also be interesting. This reviewer predicts that unlabelled-C2-Lact acts as a PS-binding competitor and causes an instantaneous detachment of labelled-C2-Lact. If this is the case, the detaching of the biosensor does not monitor the release of the cargo from the donor (because the unlabelled-C2-Lact can act as a binding competitor but not as a PS-extraction module). In contrast, if the genuine binding competitor “unlabelled-C2-Lact” exhibits an output pattern quite different from the case with Osh6, it would be evidence against the concern. In addition, the experiments with unlabelled-C2-Lact may be a good control to show that “PS deposition to acceptor LUV” can occur by the PS-transfer protein Osh6 but not by the unlabelled-C2-Lact. A similar concern occurs in the PI4P extraction experiments (Fig. 3).

We believe that the experiments with non-extractable PS, ph PS, provide clear evidence that Osh6 causes genuine extraction of the cargo. Therefore, we hope that experiments with unlabeled C2_{lact} are not necessary. Unfortunately, in the case of PI4P, we do not have the option to use a non-extractable variant. However, the calibration curves shown in Figs. 1C and 2C for C2_{lact}-CFP, and in Figs. 3C and 4C for SidC-Atto488, indicate that the PI4P biosensor, SidC-Atto488, binds PI4P even more strongly than C2_{lact}-CFP binds PS. Thus, Osh6 would not displace the biosensor even in the case of PI4P.

3) Figs. 2 and 3: How did the authors determine the “numbers of removed PS (or PI4P) per Osh6 to LUVs per minute”? Compared with the data of Fig. 2D-F, the data of Fig. 2G-I seemingly represent the plateau levels (e.g., 15-20-min point), not initial rates (e.g., 1-2-min point). Consistent with this guess, the authors noted that “the kinetics of PS extraction cannot be captured within the temporal resolution of our experiment” (L256-257), meaning it is infeasible to determine the initial rate of PS removal. Thus, the statement (L256-257) would be contradicted if “200-s points” were used for PS extraction kinetics (L573-576). Overall, the authors should clearly describe how the “numbers of transported (or extracted) lipid per Osh6 to LUVs per minute” were determined. Then, the authors should clarify whether they describe thermodynamically equilibrated states (i.e., plateau levels) or non-equilibrated transient states (i.e., kinetic rates) in the text. Due to the ambiguity of the term used in figures, it is difficult for me to assess the thermodynamic soundness of the conclusions from the part of FCCS analysis.

The reviewer is correct that we do not always present the initial rate. The reason for this is that, in the case of transport, the initial rate would be largely influenced by the extraction process, meaning the values would not accurately reflect the transfer itself. We have now included a detailed explanation of how the rate values were calculated in the Materials and Methods section.

Estimation of lipid extraction and lipid transport rates: In the case of extraction, the initial extraction rate v_{ext} was evaluated according to the following formula:

$$v_{ext} = \frac{c_{lipid(acc.)}}{c_{protein}} \times \frac{\left(\frac{G_{cc}(0)}{G_R(0)}\right)_{t=0} - \left(\frac{G_{cc}(0)}{G_R(0)}\right)_t}{\left(\frac{G_{cc}(0)}{G_R(0)}\right)_{t=0} \times t}, \quad (1)$$

where $c_{lipid(acc.)}$ and $c_{protein}$ represent the concentration of the accessible cargo lipid in the donor LUVs and of the transfer protein, respectively. $\left(\frac{G_{cc}(0)}{G_R(0)}\right)_{t=0}$ and $\left(\frac{G_{cc}(0)}{G_R(0)}\right)_t$ denote the values of the FCCS readout parameter before the transporter is added and at time t after the addition, respectively. For the fast extraction of PS, t was set to 1 minute (Figs. 1E and 6D), which is the resolution limit of our experiment. For the slower extraction of PI4P, t was set to 3 minutes (Figs. 3E and 6C) to better capture differences in kinetic decay.

In the case of the transport rate, we have evaluated the average transport rate $v_{transport}$ according to the following formula:

$$v_{transport} = \frac{(c_{lipid (acc.)} - c_{protein})}{c_{protein}} \times \frac{\left(\frac{G_{cc}(0)}{G_R(0)}\right)_{t_1} - \left(\frac{G_{cc}(0)}{G_R(0)}\right)_{t_2}}{\left(\frac{G_{cc}(0)}{G_R(0)}\right)_{t_1} \times (t_2 - t_1)}, \quad (2)$$

where t_1 is set to the time when the majority of the protein is occupied by the cargo, i.e., when the extraction process has finished. This occurs at 1 minute and 6 minutes for PS and PI4P transport, respectively. t_2 was selected to be long enough to capture differences among the individual decay processes: 10 minutes for PS (Figs. 2G, 2H, 2I) and 16 minutes for PI4P (Fig. 4G). In the formula, the accessible concentration of the lipid in the donor LUVs is adjusted to account for the fact that, once the extraction is complete, the available lipid for transport is reduced by the lipid already inside the transporter.

In Fig. 6B, the overall drop of PI4P (Δ_{PI4P}) in LUVs A during the first 10 minutes after protein addition was monitored. The drop was quantified regardless of whether it occurred through extraction or transport. We have thus modified Eq. 1 as follows:

$$\Delta_{PI4P} = \frac{c_{lipid (acc.)}}{c_{protein}} \times \frac{\left(\frac{G_{cc}(0)}{G_R(0)}\right)_{t=0} - \left(\frac{G_{cc}(0)}{G_R(0)}\right)_{t=10 \text{ min}}}{\left(\frac{G_{cc}(0)}{G_R(0)}\right)_{t=0}}. \quad (3)$$

We would also like to point out that the rates are influenced by several simplifications, such as the fact that the change in the FCCS read-out parameter is not directly proportional to the change in lipid concentration in the source LUVs. The values we calculated were intended to provide statistical context for our kinetic data, allowing us to compare the curves within individual assays and to give at least some estimate of the process rate.

The 200-second point (L573-L576) represents the measurement of the FCCS read-out prior to the protein addition, where no kinetics are expected.

4) L196-225 and Fig. 2: This reviewer has not been convinced by the main conclusion that lipid compositions of acceptor LUVs significantly affect the release of PS to the acceptor membrane. As the cell-free assay system used in this study did not input any biological energy, this assay should attain thermodynamically equilibrium states at the steady state. Thus, the data shown in Fig 2 (and also in Fig. 4 for PI4P) may be simply due to the dependence of thermodynamically equilibrated mol% of PS (and PI4P) in LUVs on co-existing lipids and membrane fluidity, not primarily on the lipid releasing speed of Osh6. The authors should address this point. Energetic and/or thermodynamic aspects of LTP-mediated lipid transfer have been described in previous reviews (*J Lipid Res*, 2018, 59, 1341; *J Cell Biol*, 2021, 220, e202012058).

We would like to thank the reviewer for this comment. Of course, the thermodynamic partitioning of the cargo between coexisting membranes plays a role, especially when fluidity is considered. Thus,

the overall effect we observe is a combination of the release rate, which is driven by specific protein-membrane interactions, and thermodynamic equilibrium, which primarily reflects lipid-membrane interactions. In summary, both interactions play a role, even in live cells with energy input. We have added the following to the text:

The effects observed in Figs. 2D-2I and 4D-4G, i.e., the impacts of membrane properties on PS and PI4P release from Osh6, respectively, reflect: (i) the thermodynamic partitioning of the cargo lipid between two coexisting membranes (the drop in $G_{cc}(0)$ at infinite time), governed by lipid-membrane interaction, and (ii) the interaction of the loaded protein with specific membrane properties (the rate of the drop before equilibrium is reached), i.e., protein-membrane interaction. In summary, the data show that transporting PS to fluid and charged membranes is advantageous. PS appears to be better accommodated in fluid membranes lacking cholesterol and in membranes bearing charge. For the release step itself, the interaction of the weakly closed lid (compared to PI4P) with acidic and fluid membranes seems to be beneficial.

The main difference between the release of PS and PI4P is that PI4P release is not enhanced by negatively charged lipids other than the cargo itself. Therefore, membrane fluidity may have a more decisive impact on its release. Here, the DOPC membrane with DAG represents the fluidity characteristics of the ER membrane, contrasting with the POPC/POPS/cholesterol membrane, which is more rigid and would more closely resemble the PM.

Minor comments

5) *The results show that PI4P in donor LUVs inhibits PS extraction (Fig. 1D, E). I agree with the explanation that PI4P serves as a binding competitor of PS for Osh6. In contrast, it is enigmatic that PI4P in acceptor LUVs inhibits PS release from Osh6 (Fig. 2D, G) because one can imagine that PI4P in the acceptor may accelerate the release of PS from Osh6 by exchanging the captured lipid ligand. Please explain this paradox briefly.*

The extraction experiments for both PS and PI4P suggest that PI4P binds more strongly to Osh6 than PS (inhibiting PI4P extraction requires a much higher excess of PS compared to inhibiting PS extraction). Additionally, simulation results and melting data for Osh6 occupied by PI4P/PS show that PI4P binds more strongly to Osh6 than PS.

According to our results, PS transport occurs spontaneously down the concentration gradient. When PI4P is present in the PS-accepting LUVs (LUVs B), PS bound to Osh6 is replaced by PI4P. The subsequent back-release of PI4P to LUVs A, which would follow the replacement, would only occur if LUVs A contain a large excess of PS, which, due to its concentration, would prevail in the Osh6 binding pocket. However, this is not the case. In the 'PS release experiment,' PS extraction and release must occur repeatedly to be detectable. If the entire capacity of Osh6 were used only once to transport PS from LUVs A to LUVs B, the resulting change in PS concentration would fall within the

biosensor's saturation range. If Osh6 binds PI4P either at the beginning or during PS exchange, PS transport is halted.

We have added to the main text:

Our extraction results confirm that PI4P is a stronger binder to Osh6 than PS. The violet squares in Figs. 1D and 3D represent the inhibition of PS extraction by PI4P and PI4P extraction by PS, respectively. A similar level of inhibition was achieved with 1 mol % of PI4P in the case of PS (1 mol %) extraction, and 20 mol % of PS in the case of PI4P (1 mol %) extraction. Consistent with the literature, molecular simulations predict a higher binding energy for PI4P compared to PS [7], and the melting point of Osh6 is also higher for PI4P than for PS [22]. This explains why PI4P release is accelerated by an excess of PS, which replaces PI4P in the Osh6 binding pocket due to its concentration. Additionally, during PS transport to PI4P-containing LUVs, the replacement of PS by PI4P halts the cyclic process of PS extraction and release, as the transporter becomes occupied by a stronger ligand and is unavailable for another cycle.

6) L502-503 “PS transport against the gradient is enabled by the negative charge of PS and by the presence of PI4P for exchange in the PS-accepting membrane”. I understand the importance of PI4P for PS transport against the gradient from the PS/PI4P-exchanging model. Please add a brief explanation from energetic or thermodynamic aspects of why the negative charge of PS is also important.

We would like to thank the reviewer for this comment, as it brings the perspective of lipid-membrane interactions back into focus. We acknowledge that we have focused solely on the interaction between the protein and the target membrane, without considering whether and how the membrane itself could stabilize the delivered cargo to prevent re-extraction.

We have added to the Discussion:

The reason that the negative charge promotes the counter-gradient transport of PS is likely not only because it facilitates lid opening, but also, from a thermodynamic perspective, due to favorable interactions between the delivered PS molecule and other charged lipids, making PS reuptake energetically unfavorable. This is supported by the kinetic decays in Figs. 2D-2F, which show that over time, the largest number of PS is transported to the charged membranes. Additionally, Fig. 6D shows that the more charged lipids the membrane contains, the less PS can be exported from it. Previous studies[26] have shown that in mixed POPC/POPS membranes, more POPS-POPS contacts are made than POPS-POPC contacts, suggesting that charged lipids attract each other, thereby stabilizing them within the membrane.

7) L42-43: I disagree with the introductory statements, which use an old review article as only one reference. Recent studies have shown that LTP-mediated non-vesicular pathways, rather than vesicular pathways for protein sorting, are predominant for inter-organelle sorting of lipids (e.g.,

reviewed in *J Lipid Res*, 2018, 59, 1341; *Nat Rev Mol Cell Biol*, 2019, 20, 85; *J Cell Biol*, 2021, 220, e202012058; *Annu Rev Cell Mol Dev* 2023, 39, 409, and references therein).

We have changed the part of the Introduction and added the citations:

The lipid molecules are transported either by vesicles that bud from the source membrane and coalesce with the target membrane [2], or by non-vesicular pathways using lipid transfer proteins (LTPs) [3-5]. It has been shown that LTPs rectify the inter organelle flux [5], are involved in organelle biogenesis, membrane repair and lipid rearrangement [6]. The bridge-like LTPs were shown to have a significant role in the lipid sorting pathways [6].

8) Regarding the lipid composition dependence on the cargo release into acceptor membranes, it would also be meaningful to touch on the possibility that an acyl chain of a specific lipid type sweeps the cargo out of the hydrophobic cavity of Osh6, as recently shown by an MD study on the CERT START domain (*J Phys Chem B* 2024: 128, 6338).

We would like to thank the reviewer for the suggestion; however, this would require all-atom MD simulations, which are lengthy and time-consuming. We have just begun such an approach for our follow-up project and are very curious to see whether it provides more insight into how the process of release or replacement occurs. At this time, however, employing this methodology would be a separate project in itself.

We have commented on that in the Results section:

Little is still known about the exact mechanism of lipid extraction and release, as it requires microsecond-long all-atomistic simulations to visualize the interaction between the membrane and the protein. The coarse-grained model used here does not allow for the observation of larger segmental mobility of Osh6. However, some work has already been done with the ceramide-transporting protein CERT [26].

9) L48: The term “phosphatidylinositol 4-phosphate (PI4P)” should be “phosphatidylinositol 4-monophosphate (PI4P)” as the authors correctly use the term “phosphatidylinositol-4,5-bisphosphate (PIP2)” in L60

We have corrected that.

Reviewer II

The experimental system used in this study provides valuable insights into the biophysical properties of lipid transport and is of significant scientific value. However, the study lacks experiments that address biological relevance. Moreover, Osh6 is a yeast protein; extending this research to homologous proteins in mammals and humans would substantially increase its impact.

We thank the reviewer for this comment. We began our work with the biophysical characterization of one of the simplest proteins from the family. In the meantime, we have established a collaboration with a lab working on yeast, and our follow-up work will certainly place our biophysical findings in a biological context. Additionally, our understanding of Osh6 is the first step towards studying human ORP8, whose ORD domain is highly homologous to Osh6.

Reviewer III

1. It is not immediately clear what new insights into the molecular mechanisms of lipid transfer are brought forward by this study. The authors emphasized the role of membrane properties in lipid transfer, such as non-cognate lipids and membrane fluidity. However, this feature is not new and has been well-demonstrated by many groups, including Drin and coworkers (e.g., Moser von Filseck, J. et al., Nat. Commun. 6, 6671, 2015). The authors seemed to suggest that strong binding between Osh6 and membranes facilitates lipid transfer. However, lipid transfer involves Osh6 unbinding from one membrane and binding to another membrane. Thus, strong membrane binding appears to hinder fast lipid transfer. Consistent with this view, many lipid binding domains in shuttle lipid transfer proteins minimally bind membranes.

We hope that the main contribution of this work to the field is the synthesis of several key factors: membrane properties (charge, fluidity), the behavior of the protein with different cargoes, and the interaction between the extracted lipid and the remaining membrane. Our aim is to use this fragmented knowledge to shed light on an important biological question: 'What enables the directionality of PS transport?' We believe we have already demonstrated that these biophysical aspects are specifically tailored to significantly contribute to the mechanism behind the counter-gradient flow of PS.

The fact that PS-loaded Osh6, compared to PI4P-loaded Osh6, prefers to interact with highly charged membranes (even though Osh6 loaded with cargo is not a strong membrane binder), that PS is more easily released to charged membranes and more difficult to extract from them, while PI4P is more easily extracted from charged membranes—;all of this together provides the mechanistic background for understanding that PS-loaded Osh6 tends to deliver its cargo to membranes rich in PS.

We have added the following comment at the beginning of the Results section to guide the reader and help them follow our motivation:

In the following section, we will examine how membrane properties, particularly the presence of charged lipids and membrane fluidity, influence the extraction and release of individual lipid cargoes. Furthermore, we will compare the membrane-binding properties of Osh6 in both its unbound form and when occupied by its ligands. Using this knowledge, we will demonstrate that Osh6 transports PS to membranes enriched in PS and containing PI4P, by effectively exchanging PS for PI4P. This provides a fundamental biophysical understanding of the processes occurring in cells.

We also clarified what we mean by Osh6 membrane binding:

LTPs, specifically their transfer domains, are designed to extract cargo when empty (i.e., bind to the membrane) and leave the membrane when loaded with cargo. Generally, these proteins are not strong membrane binders, but they do transiently interact with the membrane at some point. In the following section, we will compare the binding of empty proteins and cargo-loaded proteins to different types of membranes, with the aim of better understanding the role of charged lipids in lipid transfer.

2. Osh5 counter-transport both PS and PI4P and the concentration gradients of both PS and PI4P affect lipid transfer. The authors are encouraged to interpret their data more quantitatively based on the latest model of shuttle lipid transfer (Zhang, Y. et al. Contact, 5, 2022). Based on the lipid exchange model, the presence of a cognate lipid facilitates the exchange and transfer of another cognate lipid. Is the finding in this work consistent with this model?

We thank the reviewer for this suggestion. However, implementing such a model and fitting the decay curves with it only makes sense if the readout parameter, $G_{cc}(0)$ in this case, is directly proportional to the level of cognate lipid in the observed LUVs. Unfortunately, this is not the case for us. Generally, we can assume direct proportionality in the “low” concentration range. The reason we use our current approach, with the intention of “masking” the extraction kinetics, is that within a certain range, we can saturate the response (see the calibration curves). Additionally, the calibration curves cannot be directly used to convert the response to concentration, as they were measured in the absence of LUVs B, which also bind the biosensor. We believe we could implement the model if we put more effort into determining the exact lipid concentration at each time point of the decay, i.e., obtaining more information on the response behavior under more complex conditions. However, this goes far beyond the scope of our manuscript.

3. The manuscript should be revised to enhance readership. It is difficult to follow experimental descriptions. In the lipid extract assay, both free POPS and POPS-labeled DiD (line 578) were included in the liposome (Fig. 1A). Does Osh6 bind the dye-labeled POPS?

We would like to thank the reviewer for pointing out this unclear description. It is indeed possible to interpret it as if both labeled and non-labeled POPS were present in the system. In fact, POPS was always unlabeled, and its presence was detected using the fluorescence biosensor C2Lact-CFP.

What was labeled with DiD were LUVs A, so that we could fluorescently distinguish them from unlabeled LUVs B. DiD is a lipid tracer, an amphiphilic molecule similar to a lipid that is commonly used for membrane labeling. We apologize for the misunderstanding and have revised the experimental section to clarify this. We believe the assay schemes presented for each experiment clearly demonstrate that POPS and DiD are distinct molecules. We do not work with labeled lipids in this project, as the label has a significant impact on lipid behavior.

4. Is C2-CFP a specific sensor to PS? Note that many C2 domains broadly bind negatively charged lipids such as PI4P and PIP2.

C2_{Lact}-CFP is specific to the PS headgroup in the presence of all lipids used in this study. While it is specific, its overall response depends on the context. We have published a separate manuscript on this topic, which we also cite in the current work (Eisenreichova, 2023). Essentially, in the presence of other charged lipids or in more fluid membranes, PS is more effectively recognized by the biosensor. However, we have never observed any false recognition. We account for this when planning experiments. For instance, when investigating the effect of charged lipids, the acceptor membrane contains only 5 mol % of them.